# Verbosity ≠ Veracity: Demystify Verbosity Compensation Behavior of Large Language Models

## Abstract

Although Large Language Models (LLMs) have demonstrated their strong capabilities in various tasks, recent work has revealed LLMs also exhibit undesirable behaviors, such as hallucination and toxicity, limiting their reliability and broader adoption. In this paper, we discover an understudied type of undesirable behavior of LLMs, which we term Verbosity Compensation (VC) — similar to the hesitation behavior of humans under uncertainty — where they respond with excessive words such as repeating questions, introducing ambiguity, or providing excessive enumeration. We present the first work that defines and analyzes Verbosity Compensation, explores its causes, and proposes a simple mitigating approach. Our experiments, conducted on five datasets of knowledge and reasoning-based QA tasks with 14 newly developed LLMs, reveal three conclusions. 1) We reveal a pervasive presence of VC across all models and all datasets. Notably, GPT-4 exhibits a VC frequency of 50.40%. 2) We reveal the large performance gap between verbose and concise responses, with a notable difference of 27.61% on the Qasper dataset. We also demonstrate that this difference does not naturally diminish as LLM capability increases. Both 1) and 2) highlight the urgent need to mitigate the frequency of VC behavior and disentangle verbosity with veracity. We propose a simple yet effective cascade algorithm that replaces the verbose responses with the other model-generated responses. The results show that our approach effectively alleviates the VC of the Mistral model from 63.81% to 16.16% on the Qasper dataset. 3) We also find that verbose responses exhibit higher uncertainty across all five datasets, suggesting a strong connection between verbosity and model uncertainty. We will release our code and dataset upon acceptance.

## 1 Introduction

Large Language Models (LLMs) have demonstrated remarkable capabilities across various tasks, including reasoning (Hendrycks et al., 2021b; Cobbe et al., 2021), knowledge-based question answering (Dasigi et al., 2021; Yang et al., 2018), and planning (Valmeekam et al., 2023; Zheng et al., 2024). However, despite these impressive capabilities, numerous studies have highlighted that LLMs struggle with undesirable behaviors, such as hallucination (Huang et al., 2023), toxicity (Wen et al., 2023), and ethical bias (Tao et al., 2023), which can pose significant risks to users.

Recently, few studies have studied lengthy responses in LLMs, under chain-of-thought (Chiang & Lee, 2024; Nayab et al., 2024) and machine translation (Briakou et al., 2024) settings. However, these studies only analyze the lengthiness of the responses. Singhal et al. (2023) found the correlation between length and RLHF. However, the mechanism of decoding lengthy responses remains unknown.

In this paper, we discover an understudied undesirable behavior of LLMs. We term it Verbosity Compensation (VC) behavior. We define VC as generating responses that can be compressed without information loss when prompted to respond concisely. Instead of focusing merely on the *lengthy* issue, VC emphasizes the detailed *behavior* of generating compressible tokens with a low density of useful information. We also find VC is closely connected with the uncertainty of LLMs, demystifying the mechanism of the VC behavior, and improving the understanding of both VC and uncertainty.

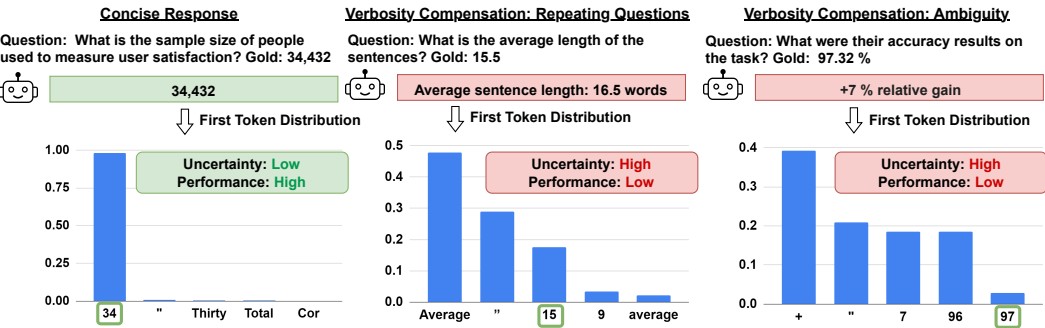

Figure 1: An illustration of comparison between concise and verbose responses. In the first response, LLM generates a concise answer, while in the second and third responses, LLM performs repeating, and ambiguity, leading to a verbose response with low performance and high uncertainty.

Interestingly, VC is similar to the hesitation behavior of humans under uncertainty (Juola, 2008; Brookshire & McNeil, 2014). Figure 1 shows a motivating example. In the first response, LLM generates a concise answer that is correct with low uncertainty. In the second and third responses, instead of generating an answer concisely, such as "16.5", LLM repeats the question, and produces ambiguity, leading to a VC response with low performance and high uncertainty. VC is harmful and undesired for both users and servers. For the users, VC will lead to confusion and inefficiency (Fowler, 1927; Oppenheimer, 2006). When an LLM enumerates multiple answers, users are unclear which one is correct. Besides, VC also leads to bias among users of different length preferences if verbose answers attain higher/lower scores. For the servers, the verbosity leads to unnecessary higher costs and higher latency by generating useless tokens.

To analyze the VC behavior systematically, we unify four long-context question-answering datasets and a reasoning-based language understanding dataset. Targeting observing indirect or low density of the information in the response rather than comparing the lengthiness of the result, we simply pick the samples with less than three tokens to easily judge VC behavior by counting the tokens in responses. We benchmark 14 newly proposed LLMs on proposed datasets. Although we find that different models and datasets exhibit diverse distribution, we can categorize VC into five distinct types, including repeating questions, enumerating, ambiguity, verbose details, and verbose format. **The result reveals a pervasive presence of verbosity compensation (VC) across all models and all datasets**. Notably, GPT-4 exhibits a VC frequency of 50.40%. Meanwhile, we found that verbose responses exhibit significantly different recall from concise ones, with a notable drop of 27.61% on the Qasper dataset, **highlighting the urgent need to disentangle verbosity with veracity**.

Next, we measure the uncertainty of model responses using perplexity and Laplacian scores for open and closed-source models. We find that verbose responses exhibit higher uncertainty across all five datasets, suggesting **a strong connection between verbosity and model uncertainty**. Finally, we leverage the connection between performance and VC to develop a routing algorithm that obtains significant improvements over the random selecting baseline and uncertainty-based routing. To mitigate VC in LLMs, we propose a simple yet effective cascade algorithm that replaces verbose responses with responses of larger LLMs. Experiments demonstrate the efficacy of the proposed algorithm through tests on three model combinations: Gemma to Gemini, Mistral to GPT-4, and Llama to Claude. The results show that our approach effectively alleviates the VC of the Mistral model from 63.81% to 16.16% on the Qasper dataset. The insights above can inspire the development of practical applications and effective mitigation strategies. Future work can mitigate the uncertainty of the LLMs by alleviating VC behavior due to the proposed connections between them.

To summarize, our contribution is threefold:

- We define VC and propose a comprehensive benchmark to evaluate 14 LLMs, revealing they suffer significantly from five types of VC.

- We conduct a rigorous analysis and connect VC to 1) model performance and 2) model uncertainty, shedding light on future applications and mitigation.

- We propose a simple but effective cascade approach to mitigate verbosity compensation in LLMs, and our extensive experiments show it is highly effective.

## 2 RELATED WORK

**Verbosity in LLM Responses**  Recently work has focused on the verbosity of LLM-generated content and its implications. Concise thoughts (Nayab et al., 2024) use prompts to constraint the length of Chain-of-thought reasoning and generate more concise responses with better performance. Ivgi et al. (2024) investigate the fallback behavior of LLM-generated responses when facing uncertainty. They found that the more advanced an LLM is, its fallback behavior shifts from sequence repetitions to degenerate text to hallucinations. Singhal et al. (2023) investigate the correlation between generated length and reinforcement learning from human feedback(RLHF) techniques, discovering that optimizing for response length is a significant factor behind RLHF. Saito et al. (2023) find that LLMs sometimes prefer more verbose answers even if they have similar qualities. Zheng et al. (2023) attack this by asking LLMs to evaluate longer and longer responses and observe if the performance increases. By contrast, Huang et al. (2024) find that GPT-4 prefers short responses in faithfulness and coverage when it comes to summarization. Unlike these works, we discover the connection between performance and verbosity compensation behavior in both CoT and general QA settings and connect verbosity to uncertainty. Besides, we use the cascading model to mitigate verbosity while they use prompt engineering.

**Uncertainty Quantification of LLMs**  With the thriving of Large Language Models (LLMs), researchers have begun exploring uncertainty quantification in LLM responses (Geng et al., 2023). For white-box models, researcher have focused on unsupervised methods including entropy (Malinin & Gales, 2020), similarity (Fomicheva et al., 2020; Lin et al., 2022), semantic (Kuhn et al., 2023; Duan et al., 2023), and logits (Kadavath et al., 2022; Chen et al., 2024), whereas for black models, the uncertainty evaluation is based on multiple sampled output of the LLMs (Malinin & Gales, 2020; Lin et al., 2023; Manakul et al., 2023) However, these works aim to improve the evaluation metrics for LLM uncertainty while we focus on connecting uncertainty with verbosity compensation behavior.

**Optimisation of LLM API Calls**  Recently, researchers have proposed to reduce the cost of leveraging a pool of LLMs (Wang et al., 2024). Some of the works train a model to predict the success rate of large or small LLMs (Ding et al., 2024) and route to the cheapest one that can succeed (Šakota et al., 2024; Lu et al., 2024). Different from routing algorithms that only pick one LLM, FragulGPT (Chen et al., 2023) use a cascade algorithm to visit LLMs from weak to strong and use an LLM evaluator to judge if the response is good enough to use (Madaan et al., 2023). Ramírez et al. (2024) leverage the uncertainty of the prediction as the evaluator to evaluate both cascading and routing structures. Similarly, Gupta et al. (2024) inspect the bias of sequential uncertainty and propose to use token-level uncertainty as criteria. Our work, by contrast, aims at mitigating verbosity compensation which has not explored before, and our evaluator is the verbosity of the response in the cascade algorithm.

## 3 VERBOSITY COMPENSATION

In this section, we first introduce the definition and quantification of verbosity compensation, and then we propose the metrics for evaluating the correlation between verbosity compensation and performance, uncertainty, and alleviating it with LLM routing.

### 3.1 VERBOSITY COMPENSATION OF LLMS

We first formalize the task as follows. A dataset $\mathcal{D}$ consists of multiple data samples where each consists of a source text $x$, a query $q$, and a ground truth $y$. A large language model $LLM(*)$ consumes the concatenation of $x, q$ and an instruction $I$ to produce the response $r$. We use $|r|$ to represent the tokens in $r$. For instruction $I$, we always ask LLM to generate as concisely as possible so that the model is instructed not to generate verbose responses. We call these tasks Concise Question Answering (CQA). Since the LLMs have maximum context window sizes $L_c$, we truncate the source to accommodate diverse context limits (details in A.3).

We define a response $r$ to exhibit verbosity if and only if it can be further compressed with fewer tokens while keeping the same meaning. To detect VC, we define verbosity compensation detector $V(x, y, r)$ (abbreviated as the verbosity detector). Using this detector, VC behavior for an LLM is defined as a triple $(x, y, r)$ where $V(x, y, r) = 1$, describing that the VC occurs in the response $r$. To quantify the frequency of VC, we define it as the ratio of VC responses in each dataset $\sum_{(x,y) \in \mathcal{D}} V(x, y, r)/|\mathcal{D}|$. It is worth noting that although $V(x, y, r)$ contains ground truth answer $y$, $y$ is usually used to provide additional information and the verbosity detectors can work reference-free.

### 3.2 PERFORMANCE AND VERBOSITY COMPENSATION

A key bias of verbosity compensation is that the performance of the verbose responses is different from the concise ones. To quantify this behavior, we propose two evaluation metrics. One is performance difference ($\Delta$), defined as the average score of the concise responses minus the average score of the verbose responses.

$$\Delta(\mathcal{D}) = \frac{\sum_{(x,y) \in \mathcal{D}} (1 - V(x, y, r)) \times \text{recall}(y, r)}{\sum_{(x,y) \in \mathcal{D}} (1 - V(x, y, r))} - \frac{\sum_{(x,y) \in \mathcal{D}} V(x, y, r) \times \text{recall}(y, r)}{\sum_{(x,y) \in \mathcal{D}} V(x, y, r)}$$

Where $r$ is the response generated by LLM and recall(y,r) is defined as $|r \cap y|/|y|$. This metric computes the difference between concise and verbose responses of a model over a dataset. If verbosity compensation has no influence on the performance, the $\Delta$ should be 0. An LLM should show zero $\Delta$ because verbosity and performance are naturally independent and thus have no relation with each other. However, if $\Delta$ is positive, then it demonstrates that verbosity responses lead to the performance drop for this model on the dataset, and vice versa. To remove the influence of the length difference between verbose and concise responses, we use recall as the scoring function. Compared with precision or F1 scores, scores are higher for verbose responses (or $\Delta$ will be smaller) because verbose responses usually contain more tokens than concise ones.

A main problem of $\Delta$ is that the recall difference between verbose and concise responses is twisted by the absolute performance of the LLMs. According to the definition, a dataset with lower performance naturally has a smaller space for performance difference. An extreme case is that the performance is zero on a dataset and the maximum $\Delta$ is zero as well. This impedes the fair comparison between datasets and models because they have diverse absolute performances. Thus, we propose relative performance difference

$$\delta(\mathcal{D}) = \Delta(\mathcal{D})/\frac{\sum_{(x,y) \in \mathcal{D}} \text{recall}(y, r)}{\sum_{(x,y) \in \mathcal{D}} 1}$$

$\delta$ can be seen as the $\Delta$ if the absolute performance of the LLMs is scaled to the same number. We use this to compare the influence of performance across datasets and LLMs.

### 3.3 VERBOSITY COMPENSATION AND UNCERTAINTY

For humans, verbosity compensation usually happens when we feel uncertain about the answers to questions. Thus, for the LLMs, it is natural to speculate verbosity compensation of LLMs is also related to the uncertainty of the model. To test this hypothesis, we evaluate the uncertainty of the LLMs with the tool proposed by Fadeeva et al. (2023). First, we split the samples according to the detector $V$ (length of response in our setting). Then, we quantify the uncertainty of each split. For open-sourced models, we use perplexity (Fomicheva et al., 2020) for evaluation, and for the close-sourced model, we use the sum of eigenvalues (Lin et al., 2023) of the graph laplacian as the metrics.

---

**Algorithm 1** Cascade Model Selection Algorithm.

**Input:** A list of LLMs $M$, A sample $(x, y, q)$, instruction $I_w$, a verbosity detector $V()$.
**Output:** A response $r$.
  order $M$ by model capability from weak to strong
  **for** LLM in $M$ **do**
    $r \leftarrow \text{LLM}(x \bigoplus q \bigoplus I_w)$
    **if** $V(x, y, r)$ is false **then**
      break
    **end if**
  **end for**
  **return** $r$

---

## 3.4 ALLEVIATING VERBOSITY COMPENSATION WITH CASCADE MODEL SELECTION

Although it is difficult to ask a single LLM to generate a concise but correct answer, the verbosity compensation behavior can be mitigated by a ensemble of multiple models. To this end, we propose a **Cas**cade Model **Sel**ection algorithm (CaSel) to increase the chance of getting concise responses. The algorithm is simple and straightforward (Algorithm 1). Given a list of LLMs from weak to strong, we first ask the weak model to generate a response with $r$ token. At any time if we detect $V(x, y, r) = 1$, we stop the generation of the current sample and redo the generation by a stronger model and repeat the process. With the power of diverse LLM, the algorithm can finally obtain a response with less verbosity and better performance.

# 4 EXPERIMENT SETUP

## 4.1 DATASET CONSTRUCTION

The principles of constructing datasets are twofold. First, the *quality* of samples needs to be high. The questions are picked from existing human-annotated datasets, with clear answers. We also filter out Yes/No, True/False, or multi-choice questions to ensure the answer cannot be simply chosen from a set of candidate answers. Second, the dataset should be *challenging* enough for LLMs with moderate performance levels. Otherwise, if the performance is close to 100 percent, the model is too certain about the answer and the phenomena is difficult to observe. Noting that most of the benchmark datasets LLMs already obtain performance higher than 90%, we craft two types of datasets to pose challenges to the model, including knowledge-based and reasoning-based question answering.

**Knowledge-based Question Answering.** This category contains the QA datasets which aim at extracting knowledge from the given source text which is long (Bai et al., 2023) or in particular position (Liu et al., 2024). Firstly, we use long-context question-answering tasks whose difficulty resides in picking out useful information across long context and gathering them to answer the question. The distractor paragraphs will also incorporate the difficulty of recognizing the needed information. Specifically, we collect the three long-form question-answering datasets as our evaluation benchmark for long-context QA. These datasets display three levels of lengths, including short (**Qasper**), medium (**LongBench**), and long (**NarrativeQA**). Qasper (Dasigi et al., 2021) is a question-answering dataset over NLP papers. It also contains extractive, abstractive, yes/no, and unanswerable questions. The average length of the source text is 4119.85 words. We also incorporate three datasets from LongBench (Bai et al., 2023) to form a new dataset. We directly name it LongBench. It include HotpotQA (Yang et al., 2018), MuSiQue (Trivedi et al., 2022), and 2WikiMultihopQA (Ho et al., 2020). The average length of the source text is 9522.36 words. NarrativeQA (Kočiský et al., 2018) is a QA dataset over entire books or movie transcripts. The answers can be abstract or extractive, yes/no, and unanswerable, and the average length is 70340.45 words.

LLMs are proven to show difficulties in understanding the information in the middle of the context (Liu et al., 2024), known as lost-in-the-middle. We pick the most challenging split of the dataset in the original work, where the gold answer is in the middle of 30 documents for a QA pair in the Natural Question dataset. We call this **NaturalQuestions_30 (NQ30).** dataset. The average length of input of NQ30 is 3602.13. More details for dataset construction can be found in Appendix A.1.

**Reasoning-based Question Answering.** We modify the multi-choice answering samples in **MMLU** (Hendrycks et al., 2021b;a) so that the options work as hints to the question. In this way, the model needs to generate the answer based on the hint rather than picking out the correct option, increasing the difficulty because of the flexibility of open-ended question answers.

For all these five datasets, to avoid the influence of gold length $y$ and the easiness of measuring the verbosity, we only pick the samples where the gold answer contains less than $n$ words. We manually inspect the samples and find that when $n = 4$, the answers are concise without descriptive context, and any answer longer than gold length is verbose. Thus, we set $n = 4$ for all datasets.

For each dataset, if the number of samples is longer than 500, we randomly pick 500 samples from them. Otherwise, we keep the entire dataset. Finally, there are 449 samples in LongBench, 410 samples in NQ30, and 500 samples in Qasper, NarrativeQA, and MMLU.

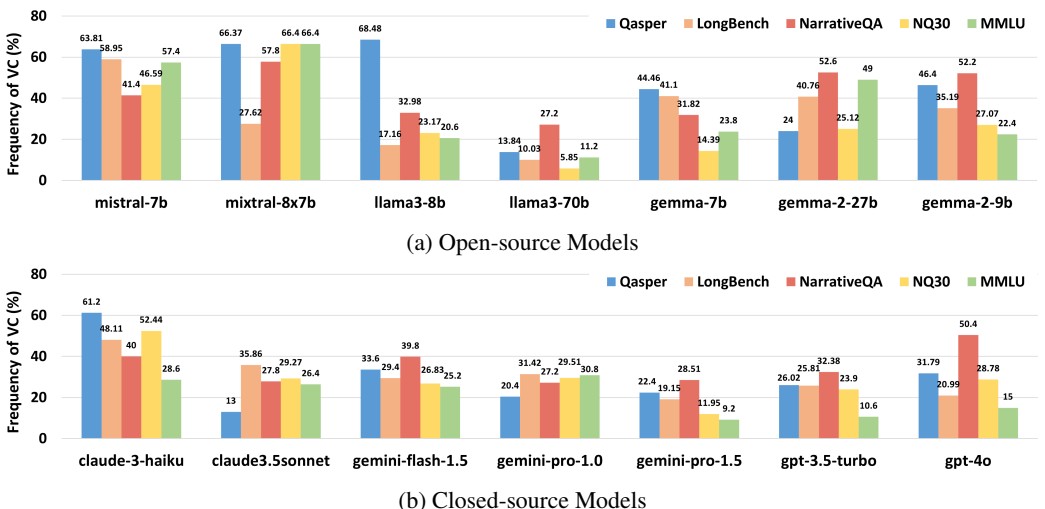

(a) Open-source Models

(b) Closed-source Models

Figure 2: Frequency of Verbosity Compensation. All models exhibit intensive verbosity compensation behavior. Among them, llama3-70b has the lowest frequency on average.

**Metrics.** We report recall when measuring verbosity compensation behavior and use F1 score for evaluation of the cascade model performance (Bai et al., 2023).

## 4.2 MODELS

We use 14 LLMs in total across all experiments including both open-source and closed-source models in 6 families: GPT, Claude, Gemini, Llama, Gemma, Mistral. Details can be found in Appendix A.2. For each model, in addition to the prompt that introduces the task, we also ask them to "generate as concisely as possible, use a single phrase if possible". Since we constraint that all samples of the datasets contain only one to three tokens, we define $V(x, y, r)$ as $|r| > 3$, meaning the number of the token in response $r$ is more than three. Also, we select 30 samples from each dataset and conduct human inspections, finding that any response that is longer can be regarded as verbose.

## 5 RESULT AND ANALAYSIS

In this section, we analyze verbosity compensation and its connection with performance and uncertainty. Then, we evaluate the proposed cascade algorithm.

### 5.1 VERBOSITY COMPENSATION

**Frequency of Verbosity Compensation Behaviors.** Figure 2 shows the frequency of each model on each dataset. As shown in the table, all the models display verbosity compensation behavior on all datasets. On average, 74.19% of the responses are verbose for mistral-7b. The best model is llama3-70b which only contains 13.62% verbose responses. Furthermore, the frequency of VC averaged on seven open-source models is 39.80% which is significantly higher than closed-source models 28.96%.

**Five Types of Verbosity Compensation Behaviors.** After showing verbosity happens frequently in LLMs, we further conduct a human annotation to inspect verbose response patterns and classify them into five types. Specifically, we choose six combinations of model and dataset (Table 1) and pick out the samples with verbose responses that are not fully correct (recall $\neq 1$, $V(x, y, r) = 1$). By checking all these samples, we classify verbosity compensation behavior into five types (Table 1): Ambiguity indicates not answering precisely; repeating question indicates repeating the tokens in the question or providing unrelated information; enumerating shows answering multiple answers in a row trying to cover the correct answer; verbose detail/format means generating more detailed explanations

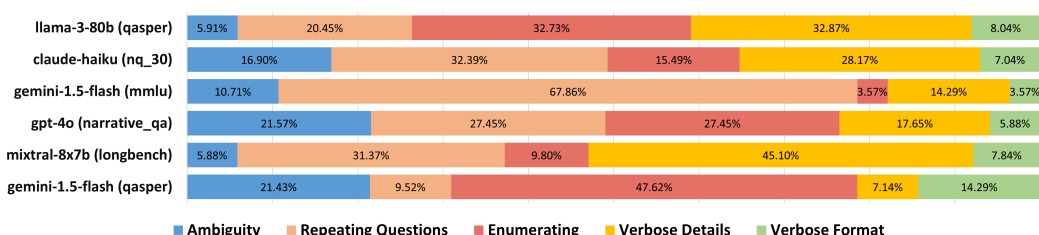

Figure 3: Human annotation of five types of verbosity compensation behavior on five datasets. Different models and datasets show diverse patterns of verbosity types.

Table 1: Examples of five verbosity compensation types.

| Dataset | Question | Gold | Model Prediction | Type |
|---|---|---|---|---|
| Qasper | What is the size of the dataset? | 3029 | It is very large | Ambiguty |
| Longbench | Which genus has more species, Dracula or Pistacia? | Dracula | Pistacia has more species | Repeat |
| NarrativeQA | What costumes are the teenagers forced to wear? | Bunny costumes | Pig , donkey , rabbit | Enumerate |
| NQ30 | who ran the fastest 40 yard dash in the nfl | Jakeem Grant | Chris Johnson 4.24 seconds | Detail |
| NarrativeQA | What types of activities occur in ...? | alleged phenomena | " Disappearances folklore " | Format |

Table 2: Overall recall comparison between verbose and concise responses. **Bold**/Underline indicate the largest positive/negative performance gap between verbose and concise responses. The verbose responses obtain a significantly different performance than the concise ones, demonstrating the strong relationship between verbosity and performance.

| | $L_c$ | Short (Qasper) | | | | Medium (LongBench) | | | | Long (NarrativeQA) | | | |
|---|---|---|---|---|---|---|---|---|---|---|---|---|---|
| | | concise | verbose | Δ | Avg. | concise | verbose | Δ | Avg. | concise | verbose | Δ | Avg. |
| mistral-7b | 8k | 66.29 | 42.85 | + 23.44 | 51.33 | 47.78 | 31.49 | +16.29 | 38.18 | 26.96 | 26.09 | + 0.88 | 25.70 |
| mixtral-8x7b | 8k | 69.60 | 48.20 | + 21.40 | 55.40 | 39.79 | 42.80 | -3.01 | 39.93 | 37.99 | 25.20 | + 12.79 | 30.60 |
| llama3-8b | 8k | 58.99 | 54.60 | + 4.39 | 55.98 | 39.71 | 28.27 | +11.44 | 37.75 | 33.11 | 15.20 | + 17.91 | 27.20 |
| llama3-70b | 8k | 55.79 | 28.19 | **+ 27.61** | 51.97 | 49.46 | 32.64 | +16.82 | 47.77 | 37.91 | 25.00 | + 12.91 | 34.40 |
| gemma-7b | 4k | 50.70 | 33.69 | + 17.01 | 43.13 | 33.28 | 15.17 | +18.11 | 25.84 | 15.82 | 3.40 | + 12.42 | 11.87 |
| gemma-2-27b | 8k | 61.89 | 40.76 | + 21.12 | 56.82 | 52.01 | 29.69 | **+22.31** | 42.91 | 43.88 | 22.43 | **+ 21.45** | 32.60 |
| gemma-2-9b | 8k | 61.94 | 49.96 | + 11.98 | 56.38 | 47.48 | 41.24 | +6.24 | 45.29 | 36.61 | 21.84 | + 14.77 | 28.90 |
| claude-3-haiku | 200k | 70.70 | 56.10 | + 14.60 | 61.77 | 48.86 | 58.01 | -9.15 | 53.26 | 55.67 | 36.25 | + 19.42 | 47.90 |
| claude-3.5-sonnet | 200k | 63.22 | 37.05 | + 26.17 | 59.82 | 59.43 | 51.97 | +7.47 | 56.76 | 55.96 | 56.12 | - 0.16 | 56.00 |
| gemini-flash-1.5 | 1m | 64.73 | 41.09 | + 23.64 | 56.79 | 59.15 | 56.06 | +3.09 | 58.24 | 36.05 | 45.48 | - 9.43 | 39.80 |
| gemini-pro-1.0 | 32k | 58.70 | 35.05 | + 23.65 | 53.87 | 47.77 | 43.07 | +4.70 | 46.29 | 24.31 | 29.41 | - 5.10 | 25.70 |
| gemini-pro-1.5 | 2m | 62.37 | 45.16 | + 17.20 | 58.51 | 62.79 | 54.07 | +8.72 | 61.12 | 37.32 | 39.29 | - 1.96 | 37.88 |
| gpt-3.5-turbo | 16k | 63.50 | 37.27 | + 26.24 | 56.68 | 51.41 | 42.51 | +8.90 | 49.11 | 41.34 | 24.85 | + 16.48 | 36.00 |
| gpt-4o | 128k | 70.22 | 43.90 | + 26.31 | 61.85 | 68.30 | 59.13 | +9.16 | 66.37 | 63.51 | 44.64 | + 18.87 | 54.00 |
| Avg. of Models | | 62.76 | 42.42 | + 20.34 | 55.74 | 50.52 | 41.87 | +8.65 | 47.77 | 39.03 | 29.66 | + 9.37 | 34.90 |

or format symbols. Then, we annotate the verbosity compensation behaviors and obtain statistics in diverse settings. As shown in Figure 3, the ratio distribution of five types of behavior varies across different models and datasets. Furthermore, the main type of Gemini-1.5-flash is repeating questions on the MMLU dataset (67.86%), and enumerating on the Qasper dataset (47.62%). In contrast, llama-3-70b mainly produces verbose details on the Qasper dataset (32.87%). This shows that different datasets or models have a significantly different distribution of the main type of verbosity behavior.

## 5.2 VERBOSITY COMPENSATION AND PERFORMANCE

**Verbose and concise responses exhibit significantly different performance.** As shown in Table 2 and Table 3, the performance difference ($\Delta \neq 0$) exists on most of the datasets and tasks, including both knowledge/reasoning-based tasks. This demonstrates that when the model performs verbosity compensation, the performance also change significantly. Among them, most of the datasets and models show lower performance on verbose samples (marked in red). For instance, llama3-70b shows 27.61% performance gap on Qasper dataset. As shown in the last column of Table 3, gemini-pro-1.0 exhibits the lowest $\Delta$ on average (7.92), and gemma-2-27b exhibits the highest (19.15). However,

Table 3: Overall recall comparison between verbose and concise responses. **Bold**/Underline indicate the largest positive/negative performance gap between verbose and concise responses. Similar to Table 2, the verbose responses obtain a significantly different performance than the concise ones.

| | $L_c$ | Lost-in-the-Middle (NQ30) | | | | MMLU (Mixed) | | | | All |
|---|---|---|---|---|---|---|---|---|---|---|
| | | concise | verbose | Δ | Avg. | concise | verbose | Δ | Avg. | Δ |
| mistral-7b | 8k | 50.76 | 39.27 | + 11.49 | 45.41 | 66.90 | 45.76 | + 21.14 | 54.77 | 14.65 |
| mixtral-8x7b | 8k | 63.74 | 51.05 | + 12.69 | 55.32 | 63.74 | 51.05 | + 12.69 | 55.32 | 11.31 |
| llama3-8b | 8k | 50.32 | 40.18 | + 10.14 | 47.97 | 58.40 | 44.82 | + 13.57 | 55.60 | 11.49 |
| llama3-70b | 8k | 51.27 | 47.92 | + 3.36 | 51.08 | 61.64 | 55.06 | + 6.58 | 60.90 | 13.46 |
| gemma-7b | 4k | 41.24 | 29.94 | + 11.30 | 39.61 | 45.60 | 44.05 | + 1.56 | 45.23 | 12.08 |
| gemma-2-27b | 8k | 57.27 | 43.85 | + 13.42 | 53.90 | 77.39 | 59.97 | + 17.42 | 68.85 | **19.15** |
| gemma-2-9b | 8k | 56.86 | 46.10 | + 10.76 | 53.94 | 68.51 | 44.49 | **+ 24.02** | 63.13 | 13.55 |
| claude-3-haiku | 200k | 60.43 | 45.12 | **+ 15.31** | 52.40 | 65.64 | 65.50 | + 0.14 | 65.60 | 8.06 |
| claude-3.5-sonnet | 200k | 55.57 | 59.03 | - 3.45 | 56.59 | 72.96 | 56.69 | + 16.27 | 68.67 | 9.26 |
| gemini-flash-1.5 | 1m | 54.50 | 48.03 | + 6.47 | 52.76 | 65.95 | 45.11 | + 20.85 | 60.70 | 8.92 |
| gemini-pro-1.0 | 32k | 50.92 | 45.87 | + 5.05 | 49.43 | 57.42 | 46.10 | + 11.31 | 53.93 | 7.92 |
| gemini-pro-1.5 | 2m | 55.10 | 43.54 | + 11.56 | 53.72 | 61.99 | 55.07 | + 6.91 | 61.35 | 8.49 |
| gpt-3.5-turbo | 16k | 53.21 | 41.33 | + 11.88 | 50.37 | 73.04 | 60.69 | + 12.35 | 71.73 | 15.17 |
| gpt-4o | 128k | 62.73 | 52.26 | + 10.47 | 59.72 | 81.25 | 68.44 | + 12.81 | 79.33 | 15.52 |
| Avg | | 54.57 | 45.25 | + 9.32 | 51.59 | 65.75 | 53.06 | + 12.69 | 61.79 | 12.07 |

*all models cannot disentangle performance with verbosity ($\Delta = 0$), highlighting the urgent need to disentangle verbosity with veracity.*

**Correlation with Model Capability.** We investigate the influence of model capability on the performance difference between verbose and concise responses $\delta$. We explore two types of model capabilities. One is general capability, represented by the arena score of ChatBot Arena (Chiang et al., 2024). We leverage the scores on the leaderboard[1](ELO) as the measurement of general capability. The other one is the capability of consuming lengthy input. For this, we investigate the influence of the size of the window context. We define the log context window size as $log(L_c/1000)$ where $L_c$ is the context window size.

Table 4: Correlation between model capability and $\delta$.

| Dataset | ELO | Log Len |
|---|---|---|
| Qasper | 0.09 | -0.26 |
| LongBench | -0.34 | -0.53 |
| NarrativeQA | -0.33 | -0.61 |
| MMLU | -0.05 | 0.13 |
| NQ14 | 0.06 | 0.02 |

Table 4 shows the correlation on five datasets. Each number in the table is computed based on the 14 data points of 14 LLMs on the corresponding dataset. As shown in the table, for Qasper, LongBench, and NarrativeQA dataset, a strong negative correlation is observed. This indicates that when modeling capability increases, the $\delta$ decreases accordingly. In contrast, for MMLU and NQ30 datasets, no obvious correlation is observed. The results show that training a stronger model will help with avoiding the influence of VC on performance for long context questions answering tasks. However, it is not helpful for MMLU and NQ30 datasets. In other words, *simply training a stronger model or extending context window cannot successfully disentangle VC and performance.*

**Verbosity compensation behavior of Chain-of-Thought reasoning.** We further conduct an experiment to demonstrate VC also happens in Chain-of-Thought (CoT) settings. To this end, we pick 100 samples from two datasets including MMLU and Qasper, and instruct the models to generate a Chain-of-thought prompt. Also, we ask the model to generate as concisely as possibility where each step contains less than 10 tokens. If any step violates this constraint, we regard this response as verbose. Thus, the verbosity evaluator $V$ is set as $\mathbb{1}\left(\bigvee_{s \in S} |s| > 10\right)$. Based on the definition, we do not restrict the number of steps of Chain-of-Thought reasoning, a short response can be verbose as well if the length of a single step is too long.

Table 5 shows the comparison between the concise and verbose responses of two models on two datasets. All settings display significant $\Delta$. For gpt-turbo-3.5, the recall gap can be as large as 24.54% on MMLU dataset. *This shows that verbosity compensation can also happen in generating longer responses, such as Chain-of-Thought reasoning samples.*

---

[1]https://lmarena.ai/

Table 5: Recall difference of Chain-of-Thought generation. Both models perform worse when they generate verbose answers, demonstrating VC also happens on CoT settings.

| | $L_c$ | Qasper | | | | MMLU | | | |
| --- | --- | --- | --- | --- | --- | --- | --- | --- | --- |
| | | concise | verbose | $\Delta$ | Avg. | concise | verbose | $\Delta$ | Avg. |
| gemma-2-9b | 8k | 35.82 | 22.73 | 13.09 | 30.12 | 60.63 | 50.00 | 10.62 | 58.42 |
| gpt-3.5-turbo | 16k | 69.05 | 47.81 | 21.24 | 61.06 | 80.95 | 56.41 | 24.54 | 68.32 |

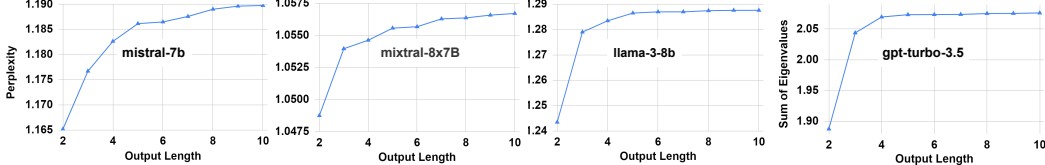

Figure 4: Uncentainty quantification of three open-sourced and one close-sourced models. The scores are averaged across all five datasets. The uncertainty increases with the increasing length of the generated output for all models.

Table 6: Frequency of Verbosity Compensation using diverse cascade models. A → B indicates combining two models using a cascade algorithm. All settings greatly reduce the frequency of VC compared with both strong and weak models.

| | Qasper | LongB | NQA | NQ30 | MMLU | Avg. |
| --- | --- | --- | --- | --- | --- | --- |
| mistral-7b | 63.81 | 58.95 | 41.40 | 46.59 | 57.40 | 74.19 |
| gpt-4o | 31.79 | 20.99 | 50.40 | 28.78 | 15.00 | 29.39 |
| mistral → gpt | **16.60** | **14.48** | **21.00** | **18.54** | **10.20** | **16.16** |
| llama3-8b | 68.48 | 17.16 | 32.98 | 23.17 | 20.60 | 32.48 |
| claude-3.5-sonnet | 13.00 | 35.86 | 27.80 | 29.27 | 26.40 | 26.47 |
| lllama → claude | **8.20** | **11.80** | **14.60** | **11.71** | **7.80** | **10.82** |
| gemma-2-9b | 46.40 | 35.19 | 52.20 | 27.07 | 22.40 | 36.65 |
| gemini-pro-1.5 | 22.40 | 19.15 | 28.51 | 11.95 | 9.20 | 18.24 |
| gemma → gemini | **15.80** | **11.14** | **18.20** | **8.29** | **4.60** | **11.61** |

## 5.3 UNCERTAINTY AND VERBOSITY COMPENSATION

**Uncertainty Evaluation.** The results are shown in Figure 4. As shown in the figure, all four models show larger uncertainty when the length of the responses increases. Especially, when the length is around three tokens, the uncertainty increases shapely. These results demonstrate that 1) when LLMs generate longer responses, they are more uncertain about the sample, and 2) *when verbosity compensation happens ($V(x, y, r) = 1$), LLMs usually are more uncertain about the sample than generating concise results*.

**Uncertainty and Length of Response $r$.** Next, we further explore the reason why the uncertainty and verbosity are connected. To achieve this, we conduct a qualitative study and plot the distribution of the softmax score of the first tokens of confident and uncertain responses in Figure 1. As can be seen, for the uncertain response, the probability distribution is more flattened, and the tokens carrying much information do not stand out (ranked high) among the candidates. The model selects the one without critical information but is safer to generate, repeating the question or being off-topic and verbose. Besides, these tokens usually cannot end a sentence grammatically, such as "Avergae" or "+", the model needs to continue generations making the response longer.

## 5.4 CASCADE MODEL SELECTION FOR MITIGATING VERBOSITY COMPENSATION

**Reducing Frequency of Verbosity Compensation.** Table 6 shows the comparison of using the proposed algorithm. As shown in the table, comparing the cascading algorithm and individual models,

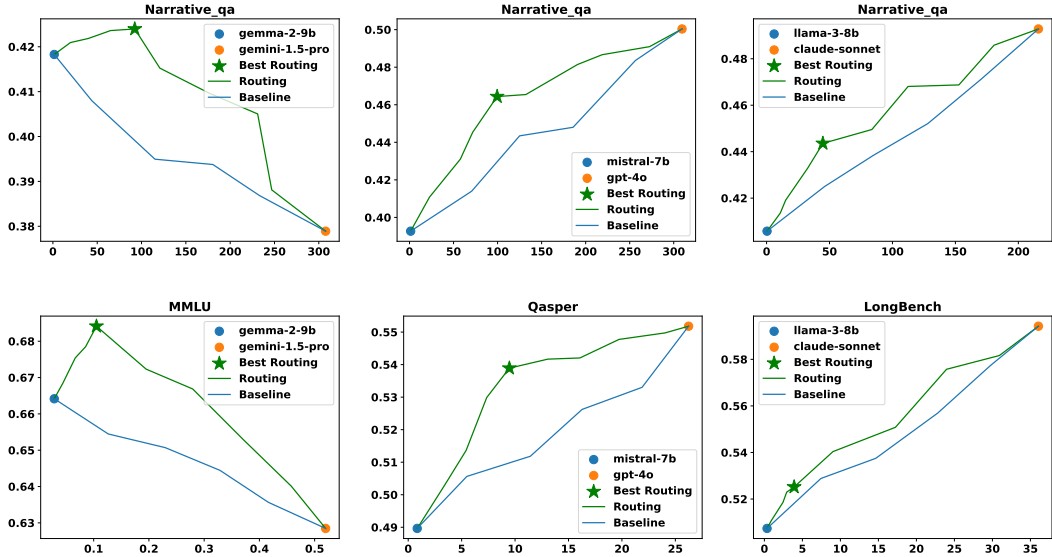

Figure 5: Routing performance of diverse models and datasets. X-axis (unit $10^{-3}$ dollars) is the average cost of running one sample. The Y-axis is the F-1 score averaged across the samples on one dataset. Routing performance (green line) is higher than the linear combination of the baseline models (blue line) with all datasets and models.

the frequency of VC decreases greatly for all settings. For instance, Mistral $\rightarrow$ GPT decreases the frequency from 63.81% (Mistral) and 31.79% (GPT) to 16.60%. It worth noting that, applying the algorithm greatly reduce the frequency of VC on both weak model and strong models.

**Using Cascade Model Selection for LLM Routing.** Model routing aims to send the sample to the proper model among a diverse collection of LLMs to generate the result so that under the same amount of API cost, the performance is better than other baselines, such as randomly choosing which model to use. We develop an LLM routing algorithm by modifying the proposed model selection algorithm. Different from model selection, we define two numbers $p_c$ and $p_v$ as the possibility of selecting a stronger model for concise and verbose responses. In this way, the cost is controllable to fulfill the diverse budget needs of users. Details are in Algorithm 3. Figure 5 shows the performance of the different datasets with three routing settings: Mistral 7b $\rightarrow$ GPT-4o, Gemma2 9b $\rightarrow$ Gemini-1.5-pro, and LLaMA-3-8b $\rightarrow$ Claude-3.5-sonnet. We run each $p_v, p_c$ setting ten times and compute the average to obtain the green lines and we run ten times that we randomly choose a weaker or stronger model with different probability to draw the blue line serving as the baseline. Specifically, for the stars in each figure, $p_v = 1$ and $p_c = 0$, degenerate to the proposed model selection algorithm. As shown, the performance of routing is better than the baselines for all models, datasets, and settings. Furthermore, the routing results from Gemma-2 to Gemini-1.5 are better than the individual performance of both models. This indicates that *the routing algorithm improves the performance for all settings and can surpass the performance of stronger models with less cost.*

## 6 CONCLUSION

In this paper, we systematically analyze the verbosity compensation behavior of LLM responses. We first classify verbosity into five types and find all the models display high frequency on verbosity responses. We further explore the reason behind and find uncertainty highly likely connected to the phenomenon. We also propose a cascade model selection algorithm to mitigate it. Intensive experiments show that LLMs suffer from verbosity compensation and the proposed simple approach mitigates the verbosity compensation effectively.

## REPRODUCIBILITY STATEMENT

We will provide a GitHub repository that includes the dataset of five datasets, code for running 14 LLMs, and the predicted results for the use of studies. We also include the prompt, settings, and implementation details in appendix for the reproduction. The tools for evaluating uncertainty is open-sourced, thus can be used freely as well.

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

## A IMPLEMENTATION DETAILS

### A.1 DETAILS OF DATASET CONSTRUCTION

Firstly, we use long-context question-answering tasks whose difficulty resides in picking out useful information across long context and gathering them to answer the question. The distractor paragraphs will also incorporate the difficulty of recognizing the needed information. **Qasper (short)** (Dasigi et al., 2021) is a question answering dataset over NLP papers. It also contains extractive, abstractive, yes/no, and unanswerable questions. The average length of the source text is 4119.85 words. We also incorporate three datasets from **LongBench (medium)** (Bai et al., 2023) to form a new datasets. HotpotQA (Yang et al., 2018) is a Wikipedia-based multi-hop QA dataset. It requires reasoning across multiple passages to find the answer. MuSiQue (Trivedi et al., 2022) is a multi-hop QA dataset. It is much more difficult than HotpotQA as it contains more hops in one sample, unanswerable questions, and harder distracting content. 2WikiMultihopQA (Ho et al., 2020) It consists of up to 5-hop questions that are synthesized through manually designed templates. The average length of the source text is 9522.36 words. Different from the previous datasets, **NarrativeQA (long)** (Kočiský et al., 2018) is a QA dataset over entire books or movie transcripts. The answers can be abstract or extractive, yes/no, and unanswerable, and the average length is 70340.45 words. Both Qasper and NarrativeQA datasets in our benchmark are extracted from SCROLLS dataset (Shaham et al., 2022). **NaturalQuestions_30 (NQ30).** LLMs are proven to show difficulties in understanding the information in the middle of the context (Liu et al., 2024), known as lost-in-the-middle. We pick the most challenging split of the dataset in the original work, where the gold answer is in the middle of 30 documents for a QA pair in the Natural Question dataset. We call this NQ30 dataset. The average length of input of NQ30 is 3602.13.

### A.2 DETAILS OF LARGE LANGUAGE MODELS

We include 2 models from Mistral AI[2], among them, **mistral-7b** is its first proposed dense model while **mixtral-8x7b** enhances the 7b model by incorporating a sparse mixture of experts. Gemini (Team et al., 2023; Reid et al., 2024) is a family of LLMs proposed by Google from which three versions of LLMs are selected, including **gemini-pro-1.0**, **gemini-flash-1.5**, and **gemini-flash-1.5**. Built from the research and technology used to create Gemini models, Gemma (Team et al., 2024a;b) is a family of lightweight, open models. We include **gemma-7b**, **gemma-2-9b**, and **gemma-2-27b** for experiments. LlaMA 3 (Dubey et al., 2024) is a family of LLMs with dense Transformer structure. We include **llama-3-8b** and **llama-3-70b** for experiments. Claude (Anthropic, 2024) is a family of large language models developed by Anthropic. We include two models in ascending order of capability: **claude-3-haiku**, **claude-3.5-sonnet**. We also include two versions of GPT models[3], including **gpt-3.5-turbo** and **gpt-4o** in experiments.

During experiments, we use the default parameters of all 14 LLMs. We run gemma, llama, and mistral models from Huggingface[4] on 8 A100 GPUs. For gpt, claude, and gemini models, we run with the official API from the official website. For all datasets, we use the same prompt shown in Table 7. We design a reinforced prompt to ensure LLM understands concise responses are required. Thus, we reinforce the prompt by repetition, and explanation, especially for the weaker models, making a fairer comparison by avoiding failing to understand instructions. We evaluate the robustness of VC against diverse prompts in Apendix C.3.

### A.3 INPUT CHUNKING ALGORITHM

Before we feed the input into the model, we first chunk the source so that the model can consume it. As shown in Algorithm 2, we first split the source into sentences and feed as many sentences as possible to LLMs.

---

[2]https://docs.mistral.ai/getting-started/models/
[3]https://openai.com/
[4]https://huggingface.co/

Table 7: Prompt of all models on all datasets.

You are given an article and a question.
Answer the question as concisely as you can, using a single phrase if possible. Article:
{Source Documents}
Question:
{Question $q$}
Using a single phrase rather than a sentence. Please answer in 3 words.
Do not repeat any question-related information or explain the answer.
The answer is:

Table 8: The full name and the cost of tokens for each model. The unit of input/output cost is dollar per one million tokens.

|  | Input Cost | Output Cost | Model Full Name |
| --- | --- | --- | --- |
| mistral-7b | 0.17 | 0.2 | mistralai/Mistral-7B-Instruct-v0.3 |
| mixtral-8x7b | 0.24 | 0.24 | mistralai/Mixtral-8x7B-Instruct-v0.1 |
| llama3-8b | 0.05 | 0.08 | meta-llama/Meta-Llama-3-8B-Instruct |
| llama3-70b | 0.59 | 0.79 | meta-llama/Meta-Llama-3-70B-Instruct |
| gemma-7b | 0.07 | 0.07 | google/gemma-7b-it |
| gemma-2-27b | 0.8 | 0.8 | googlegemma-2-27b-it |
| gemma-2-9b | 0.2 | 0.2 | google/gemma-2-9b-it |
| claude-3-haiku | 0.25 | 1.25 | claude-3-haiku-20240307 |
| claude-3.5-sonnet | 3 | 15 | claude-3-5-sonnet-20240620 |
| gemini-flash-1.5 | 0.35 | 1.05 | gemini-1.5-flash |
| gemini-pro-1.0 | 0.5 | 1.5 | gemini-1.0-pro |
| gemini-pro-1.5 | 3.5 | 10.5 | gemini-1.5-pro |
| gpt-3.5-turbo | 0.5 | 1.5 | gpt-3.5-turbo-0125 |
| gpt-4o | 5 | 15 | gpt-4o-2024-05-13 |

---

**Algorithm 2** Input Chunking Algorithm.

---

**Input:** Source input $x$, query $q$, LLM window size $k$, instruction $I_w$.
**Output:** A chunk $c$ that LLM can consume.
  Split the source $x$ into sentences $\{s_1, s_2, \cdots, s_n\}$
  Initialize $c \leftarrow$ empty string
  Initialize length budgets $B \leftarrow k - \text{count\_token}(q) - \text{count\_token}(I_w)$.
  **for** $s$ in $s_1, s_2, \cdots, s_n$ **do**
    **if** count_token(c) + count_token(s) ¿ B **then**
      break
    **end if**
    $c \leftarrow c \bigoplus s$   // $\bigoplus$ indicates concatenating two strings with a blank.
  **end for**
  **return** c

---

## A.4 LLM ROUTING ALGORITHM

Algorithm 3 shows the pseudo-code of LLM Routing. Different from the cascade algorithm for mitigating VC, this algorithm contains two probabilities that are used to control the budget of a single call. The algorithm mimics the real cost by counting tokens in the input and output, timing by the cost per token. We collect the cost of each model from website[5] and use it collected cost to ensure the fairness of comparison. The full name of all models and the price we use in LLM routing algorithm is shown in Table 8.

---

[5]https://artificialanalysis.ai/models

Table 9: Frequency of Verbosity Compensation. All models have verbosity compensation behavior. Among them, llama3-70b has the lowest frequency on average.

| | $L$ | Qasper | LongB | NQA | NQ30 | MMLU | Avg. |
|---|---|---|---|---|---|---|---|
| mistral-7b | 8k | 63.81 | 58.95 | 14.20 | 46.59 | 57.40 | 74.19 |
| mixtral-8x7b | 8k | 66.37 | **4.38** | 57.80 | 66.40 | 66.40 | 52.27 |
| llama3-8b | 8k | 68.48 | 17.16 | 32.98 | 23.17 | 20.60 | 32.48 |
| llama3-70b | 8k | 13.84 | 10.03 | 27.20 | **5.85** | 11.20 | **13.62** |
| gemma-7b | 4k | 44.46 | 41.10 | 31.82 | 14.39 | 23.80 | 31.11 |
| gemma-2-27b | 8k | 24.00 | 40.76 | 52.60 | 25.12 | 49.00 | 38.30 |
| gemma-2-9b | 8k | 46.40 | 35.19 | 52.20 | 27.07 | 22.40 | 36.65 |
| claude-3-haiku | 200k | 61.20 | 48.11 | 40.00 | 52.44 | 28.60 | 46.07 |
| claude-3.5-sonnet | 200k | **13.00** | 35.86 | 27.80 | 29.27 | 26.40 | 26.47 |
| gemini-flash-1.5 | 1m | 33.60 | 29.40 | 39.80 | 26.83 | 25.20 | 30.97 |
| gemini-pro-1.0 | 32k | 20.40 | 31.42 | 27.20 | 29.51 | 30.80 | 27.87 |
| gemini-pro-1.5 | 2m | 22.40 | 19.15 | 28.51 | 11.95 | **9.20** | 18.24 |
| gpt-3.5-turbo | 16k | 26.02 | 25.81 | 32.38 | 23.90 | 10.60 | 23.74 |
| gpt-4o | 128k | 31.79 | 20.99 | 50.40 | 28.78 | 15.00 | 29.39 |
| Avg | | 34.53 | 31.71 | 44.11 | 31.98 | 31.14 | 34.69 |

---

**Algorithm 3** Cascade Model Selection Algorithm for LLM Routing.

---

**Input:** A list of LLMs $M$, A sample $(x, y, q)$, instruction $I_w$, a verbosity detector $V()$, possibility for routing on concise responses $p_c$, possibility for routing on verbose responses $p_v$.
**Output:** A response $r$.
   order $M$ by model capability from weak to strong
   Set $p_c$ to 1 if $p_v \neq 1$ {We ensure routing on verbose responses first.}
   **for** LLM in $M$ **do**
      $r \leftarrow$ LLM$(x \bigoplus q \bigoplus I_w)$
      **if** $V(x, y, r)$ is false **then**
         $prob \leftarrow$ A random number from 0 to 1
         **if** $prob \geq p_c$ **then**
            break {Do not route for concise responses with $1 - p_c$ probability}
         **end if**
      **else**
         $prob \leftarrow$ A random number from 0 to 1
         **if** $prob \geq p_v$ **then**
            break {Do not route for verbose responses with $1 - p_v$ probability}
         **end if**
      **end if**
   **end for**
   **return** $r$

---

# B DETAILS OF EXPERIMENTAL RESULTS

## B.1 FREQUENCY OF VERBOSITY COMPENSATION

Table 9 shows the detail numbers of frequency of verbosity compensation behavior.

## B.2 UNCERTAINTY VERSES LENGTH

Table 10 shows some examples of verbose and concise responses and the distribution of the first token.

## B.3 MODEL CAPABILITY AND RELATIVE DELTA

Figure 6 plots the Correlation between model window size and $\delta$, visualizing the negative correlation score in Table 4. The models with the stronger capability to consume lengthy input obtain lower relative delta, indicating verbosity compensation is better avoided. Also, the decreasing speed of the tendency line ranks as follows: Long (NarrativeQA), Medium (LongBench), and Short (Qasper).

Table 10: Distribution of the first token generated by llama-3-8b on Qasper dataset. When generating the verbose responses, the model is uncertain about the gold answer. Thus, starting with some commonly seen tokens. In contrast, the concise responses directly start with gold answers with high confidence.

| Three Examples of Verbose Responses | |
|---|---|
| Question | What is the average length of the sentences? |
| Gold | 15.5 |
| Vanilla | Average sentence length: 16.5 words |
| Distribution | [['Average', 0.47828879952430725], ['', 0.29009681940078735], ['15', 0.17595264315605164], ['9', 0.034647125750780106], ['average', 0.021014543250203133]] |
| Question | What were their accuracy results on the task? |
| Gold | 97.32 % |
| Vanilla | +7 % relative gain |
| Distribution | [['+', 0.39178282022476196], ['', 0.20970624685287476], ['7', 0.1850651204586029], ['96', 0.1850651204586029], ['97', 0.028380654752254486]] |
| Question | What's the size of the previous largest OpenIE dataset? |
| Gold | 3,200 sentences |
| Vanilla | " 13 times larger " |
| Distribution | [['', 0.5911211967468262], ['13', 0.3585331439971924], ['"L', 0.02022705413401127], ['Three', 0.01785031333565712], ['3', 0.012268329039216042]] |
| Three Examples of Concise Responses | |
| Question | What is the sample size of people used to measure user satisfaction? |
| Gold | 34,432 |
| Vanilla | 34,432 users |
| Distribution | [['34', 0.9797704219818115], ['', 0.008476674556732178], ['Thirty', 0.006601640954613686], ['Total', 0.004004097543656826], ['Cor', 0.0011471931356936693]] |
| Question | Which news organisations are the headlines sourced from? |
| Gold | BBC and CNN |
| Vanilla | BBC and CNN |
| Distribution | [['BBC', 0.9247239232063293], ['', 0.04062953218817711], ['"B', 0.0279242433360757828], ['B', 0.003779135411605239], ['"C', 0.00294319353399770737]] |
| Question | which datasets did they experiment with? |
| Gold | Europarl MultiUN |
| Vanilla | Europarl MultiUN |
| Distribution | [['Eu', 0.9808066487312317], ['Euro', 0.009615491144359112], [' Europ', 0.0074885510839521885], ['', 0.0014745831722393632], ['European', 0.000614697695709765]] |

This means that the effectiveness of the length capability on disentangling verbosity and performance is more significant when the task has a longer input.

### B.4 TRUNCATION PRINCIPLE

We conducted an experiment on Qasper dataset with llama-3-8b and found that When the response is verbose, only keep the first 4 tokens, then stop the generation. The recall only drops from 44.93% to 43.13%. In other words, if the gold answer is not in the first 4 tokens, then the model is not likely to generate it in the rest of the tokens.

## C SUPPLEMENTARY EXPERIMENTS

### C.1 COMPARISON WITH UNCERTAINTY-BASED ROUTING ALGORITHM

We further conduct an analysis to compare the performance of the proposed routing algorithm with the uncertainty-based routing algorithm in addition to the random baselines. For the uncertainty-based routing algorithm, we first use perplexity as the metric to rank the uncertainty of the responses generated by a small model. We select top K% uncertain samples and replace them with the responses generated by the larger model. We select K from a set of $\{0, 10, 20, \cdots, 100\}$ and connect them to draw the curve in Figure 7. As can be seen, although the uncertainty-based routing algorithm can obtain a better performance than the random baseline, it is still worse than the proposed algorithm

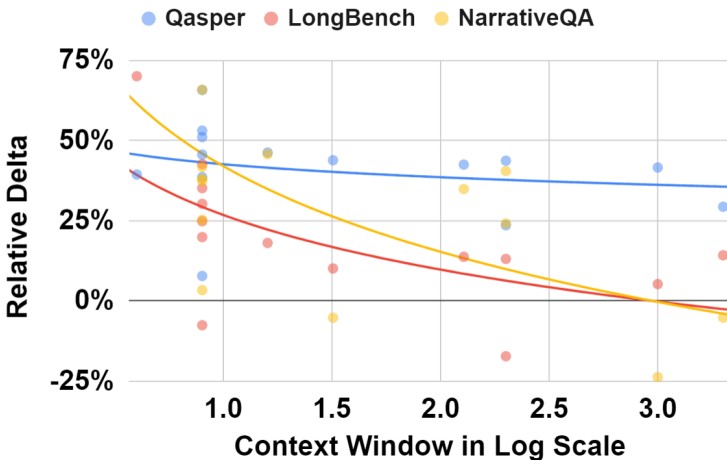

Figure 6: Correlation between model window size and $\delta$. Results show that the model with a longer context window shows less $\delta$ on Qasper, LongBench, and NarrativeQA dataset.

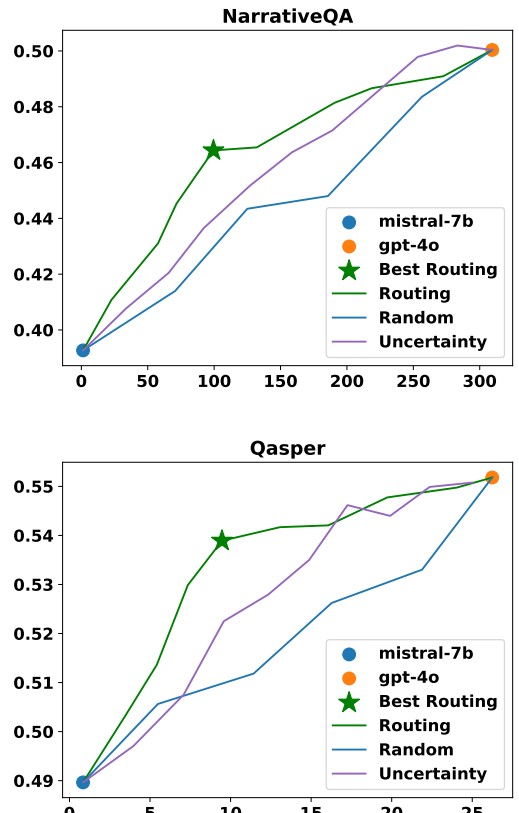

Figure 7: Routing performance of Mistral-7b to GPT-4o. X-axis (unit $10^{-3}$ dollars) is the average cost of running one sample. The Y-axis is the F-1 score averaged across the samples on one dataset. Routing performance (green line) is higher than the random baseline models (blue line) and uncertainty-based baseline (purple .

by comparing the AUC of the figure (Area Under the Curve), demonstrating the effectiveness of the proposed algorithm.

## C.2 VERBOSITY COMPENSATION IN TRIP PLANNING DATASET

To further demonstrate that VC generally occurs in diverse open-ended tasks with diverse response lengths, we run a trip planning dataset from the Natural-Plan benchmark Zheng et al. (2024) using two Llama-3 models and test VC frequency and performance gaps. The task is to find the itinerary regarding the order of visiting N cities. We randomly select 500 data points from the dataset to form our dataset. For the prompt design, we follow the zero-shot prompt in the original paper and add one sentence "Answer as concisely as possible, each step contains less than 10 words". For the verbosity detector follows our CoT setting: $V(x, y, r) = \mathbb{1}\left(\bigvee_{s \in S} |s| > 10\right)$. The results are shown in Table 11. VC also occurs frequently in trip planning, demonstrating the general presence of VC in both short- and long-response open-ended tasks.

Table 11: VC frequency and performance gaps on trip planning dataset.

|  | concise | verbose | $\Delta$ | Avg. | VC Freq. |
|---|---|---|---|---|---|
| llama-3-8b | 15.18 | 3.62 | 11.56 | 9.22 | **51.49** |
| llama-3-70b | 21.81 | 4.87 | **16.94** | 19.63 | 12.87 |

## C.3 ROBUSTNESS OF VERBOSITY COMPENSATION AGAINST PROMPT CHOICES

As shown in Table 7 We design a reinforced prompt to ensure LLM understands concise responses are required. Thus, we reinforce the prompt by repetition, explanation, etc., especially for the weaker models, making a fairer comparison by avoiding failing to understand instructions.

We further experiment with multiple possible prompts to show VC is not overfitting to certain prompt settings. We aim to show that as long as the model knows to generate as concise as possible, we can observe significant VC behaviors.

Table 12 shows the performance gap on MMLU and Qasper datasets using Llama-3-8b with different prompt designs. As can be seen, compared with the original prompt, the variation of the prompt can also observe a significant $\Delta$ over both datasets. This demonstrates the robustness of VC against the choice of prompts. It is worth noting that, "Answer as concise as possible" yields the highest scores on two datasets, as well as the highest $\Delta$, demonstrating a simpler prompt with less constraint might generate a larger performance gap between concise and verbose responses.

Table 12: Comparison between original and other variations of the prompts. VC consistently occurs, demonstrating the robustness of the VC against prompts.

|  | MMLU | | | | Qasper | | | |
|---|---|---|---|---|---|---|---|---|
|  | concise | verbose | $\Delta$ | Avg. | concise | verbose | $\Delta$ | Avg. |
| *Prompt in Table 7* | | | | | | | | |
|  | 58.4 | 44.82 | 13.57 | 55.6 | 58.99 | 54.6 | 4.39 | 55.98 |
| *Using a single phrase rather than a sentence. Please answer in 3 words.* | | | | | | | | |
|  | 55.13 | 43.43 | 11.71 | 52.70 | 54.22 | 48.11 | 6.11 | 51.30 |
| *Answer as concise as possible.* | | | | | | | | |
|  | 68.04 | 50.26 | **17.78** | 61.07 | 70.17 | 60.44 | **9.73** | 63.63 |

## C.4 EVALUATION OF VERBOSITY AND PERFORMANCE ON SAME TEST INSTANCES

As shown in Table 2, and Table 3, the performance of concise and verbose samples is based on the split of the dataset. There is no overlap between the samples in the concise and verbose split. To prevent the influence of bias in different instances, we conduct an analysis that fixes the test instances and compares different models. Specifically, for each instance, we calculated the ratio of LLMs exhibiting VC behavior and reported the averaged ratio across datasets in Table 13. This approach also increases the robustness of our findings, as the support (number of samples) for each dataset is

14 times higher than when using a single model. As shown in the table, the performance $\delta$ is still pervasive for all five datasets. Specifically, on the Qasper dataset, the $\Delta$ reaches 16.22%

Table 13: Overall recall comparison between verbose and concise responses. Each dataset contains the prediction from all 14 LLMs.

|  | concise | | verbose | | overall | | |
| --- | --- | --- | --- | --- | --- | --- | --- |
|  | Recall | Support | Recall | Support | $\Delta$ | VC Freq. | Avg. Recall |
| Qasper | 61.85 | 2272 | 45.63 | 389 | **16.22** | 32.46 | 56.59 |
| LongBench | 50.31 | 1912 | 44.22 | 375 | 6.10 | 30.42 | 48.46 |
| NarrativeQA | 38.09 | 2540 | 31.67 | 355 | 6.42 | **36.29** | 35.76 |
| MMLU | 65.09 | 1694 | 51.47 | 475 | 13.62 | 24.20 | **61.79** |
| NQ30 | 53.34 | 1516 | 44.89 | 362 | 8.45 | 26.41 | 51.10 |

## C.5 LATENCY COMPARISON OF CASEL ALGORITHM AND INDIVIDULE MODELS

We conduct an analysis to compare the useless token generated and the time cost of individual models and the CaSel algorithm on two datasets using Mistral-7b and GPT-4o. To assess the number of useless tokens generated, given a response $r$, we first define the useless tokens as the part with longer than 3 tokens in response $r$: $\sum_{i=1}^{N} \max(0, |r_i| - 3)$, where $N$ is the number of samples in a dataset. As shown in Table 14, with our proposed cascade algorithm, the total inference time might be higher than using a small model (0.79 vs. 1.21 seconds per sample) and lower than using a large model (14.86 vs. 5.93 seconds per sample), but the number of useless tokens generated is much less. On the other hand, by using the proposed algorithm, the useless tokens generated decrease from 596/327 to 93, mitigating the VC rate from 41.40% to 21.00% on the NarrativeQA dataset, demonstrating that useless tokens greatly decrease by using the proposed algorithm.

## C.6 THE INFLUENCE OF THE DIGITS IN RESPONSES

We analyze the performance and VC frequency of the samples with and without numbers using llama-3-8b on the Qasper and NarrativeQA dataset. The results are shown in Table 15. Although the model is easier to perform better on the sample without numbers, the VC frequency is relatively lower for the responses with digits. To understand the reason, we further inspect the Qasper dataset, we find that the samples with numbers are not as open-ended as the ones without numbers, meaning that the search space of the answers with numbers is smaller. This leads to a lower VC frequency and is easier to answer.

## C.7 RESPONSE LENGTH OF CHAIN-OF-THOUGHT EXPERIMENTS

Our evaluation is not limited to gold answers with less than 4 words. To demonstrate the generalization of the proposed VC behavior, we run the experiments on Chain-of-Though settings where the responses can contain more than 300 words. Table 16 shows the statistics of Chain-of-Thought

Table 14: Comparison of the number of generated useless tokens and inference time. # Mistral/GPT indicates the number of useless tokens generated by Mistral-7b and GPT-4o on the dataset. # Total is the sum of # Mistal/GPT, showing the total number of useless tokens. Infer. Time is the running time of the algorithm per sample (Unit: second). CaSel (Mistral → GPT) generated the fewest number of useless tokens and maintained the lowest VC frequency. The inference time is higher than the small model but still lower than the larger model.

|  | Qasper | | | | | NarrativeQA | | | | |
| --- | --- | --- | --- | --- | --- | --- | --- | --- | --- | --- |
|  | # Mistral | # GPT | # Total | VC Freq. | Infer. Time | # Mistral | # GPT | # Total | VC Freq. | Infer. Time |
| Mistral-7b | 663 | N/A | 663 | 63.81 | **0.80** | 596 | N/A | 596 | 41.40 | **1.22** |
| GPT-4o | N/A | 207 | 207 | 31.79 | 1.27 | N/A | 327 | 327 | 50.40 | 14.86 |
| Mistral → GPT | 0 | 86 | **86** | **16.60** | 1.21 | 0 | 93 | **93** | **21.00** | 5.93 |

Table 15: Comparison between responses with digits and without digits. The responses with digits show lower verbosity compensation frequency.

|            | Qasper  |         |           |          | NarrativeQA |         |          |          |
|------------|---------|---------|-----------|----------|-------------|---------|----------|----------|
|            | concise | verbose | Avg.      | VC Freq. | concise     | verbose | Avg.     | VC Freq. |
| w/o digits | 58.99   | 53.66   | **56.18** | 52.63    | 33.39       | 18.18   | **27.21**| 40.66    |
| w/ digits  | 58.97   | 57.73   | 58.40     | **45.83**| 56.25       | 10.00   | 38.46    | **38.46**|

Table 16: Lengths of the generated responses under chain-of-thought setting. The maximum length of the generated results can reach more than 300 words demonstrating that VC occurs in long response settings.

|              | MMLU     |          |          |          | Qasper   |          |          |          |
|--------------|----------|----------|----------|----------|----------|----------|----------|----------|
|              | VC Freq. | Min Len. | Max Len. | Avg Len. | VC Freq. | Min Len. | Max Len. | Avg Len. |
| gpt-3.5-turbo| 51.49    | 3        | 90       | 26.24    | 37.62    | 4        | 81       | 23.38    |
| gemma-2-9b   | 20.79    | 9        | 107      | 27.92    | 43.56    | 18       | 103      | 37.08    |
| llama-3-8b   | 43.56    | 15       | 333      | 57.14    | 44.15    | 20       | 185      | 50.15    |

experiments. The average response length can reach more than 50 words, and the VC behavior is still pervasive.

