# OpenReview forum: "Verbosity $\neq$ Veracity: Demystify Verbosity Compensation Behavior of Large Language Models"
_ICLR.cc/2025/Conference — Submitted to ICLR 2025_

### Official Review · Reviewer_esHV · 2024-10-30

**Soundness:** 2
**Presentation:** 3
**Contribution:** 2
**Rating:** 5
**Confidence:** 4

**Summary:**

The paper introduces a novel concept of verbosity compensation in LLMs. They define this term, explore it's presence in different models using combination of different datasets, also they try to come up an algorithm to tackle this issue of VC. The paper defines verbosity compensation as a tendency of LLMs to generate overly verbose responses when the LLMs are unsure about the answer. Coming to the evaluation part there are different sections of it with combination of automated evaluation and manual human evaluations. The paper does have some interesting things but there are some issues with the actual implementation and also writing, it is not very clear the exact research question the paper is trying to solve.

**Strengths:**

The model has a new concept of verbosity compensation

They explore 14 different LLM for comparison and performance evaluation

The idea of different type of VCs

The work also introduces algorithm to mitigate the issue that has been defined in the paper

**Weaknesses:**

- > The problem doesn't seem to very well defined, I couldn't understand the exact research question the paper is trying to solve.
If the problem is about the length of output, I don't see it as a major issue.

 -> The dataset used for this experiment is creates using existing datasets from previous works, most of these are from previous years and there is high chance that these were in the training data that was used for training different LLMs used in the paper and the details of these data-leakage are not explored or mentioned.

-> The prompt used for the experiment is not clear, more details on why 3 words was choose and why is it mentioned multiple times single phrase if possible, but then you mention single phrase and also 3 words. Given that the work depends a lot on the prompt, more prompts should have been experimented and more details of those results are needed.

-> Though there is mention of verbosity detector the actual definition and formula on how you calculate it is missing.

-> There is introduction of algorithm, the paper mentions that VC might causes latency, there can be latency due to algorithm as well.

-> Writing can be improved

-> Most of the examples mentioned in the paper have some numbers in the answers, out of 3 correctly given answers 2 doesn't have any numbers, given that there is existing research that LLMs are not good at the numbers, it would be better to compare the behavior on those lines as well, if the output is wrong because of the numbers and not other factors.

**Questions:**

N/A

---

> ### Author Response · Authors · 2024-11-27
> **Author Response (Part 1)**
>
> We thank the reviewer for the helpful feedback! The constructive questions are very helpful for improving our paper. We address each of your questions below. We also submitted a newer PDF that addresses your comments and rewrites the abstract and introduction with a clearer statement of the research problem.
>
> ---
>
> > ### **Weakness 1**
> > **The problem doesn't seem to very well defined, I couldn't understand the exact research question the paper is trying to solve. If the problem is about the length of output, I don't see it as a major issue.**
>
> Our research does not aim to address the lengthiness of responses itself. The conventional understanding of verbosity primarily focuses on the **lengthiness** of responses. In contrast, our work highlights the **undesired indirect or hesitant responses** of Verbosity Compensation exhibited by LLMs. This behavior goes beyond mere length and includes tendencies such as repeating questions, introducing ambiguity, or providing excessive enumeration when tasked with generating concise answers.  While we use response **length** as a simple yet effective **tool** to identify such behavior, the core contribution of our work is independent of length. To clearly differentiate from the lengthy issue, our proposed Verbosity Compensation specifically emphasizes an *undesired behavior of not generating a valid answer to the question directly and decisively*.
>
>
> For instance, when given the question:
>
> What is the capital city of the United States? Answer as concisely as possible:
>
> Some decisive answers (no VC behavior):
> - Washington DC (Correct and concise)
> - Not sure (Confessing and concise)
> - Paris (Wrong but concise)
>
> However, LLMs tend to answer indirectly or hesitantly (VC behavior):
> - The capital city of the US is Paris (Repeating Question)
> - A city in a certain part of the North America (Ambiguity)
> - New York City, Paris, or San Francisco (Enumerating)
>
> Also in this example, responses with VC are not too lengthy, showcasing that VC is not simply a lengthy issue.
>
> While LLMs may occasionally provide incorrect answers, we expect their responses to remain concise and decisive. We define this undesired behavior as verbosity compensation and argue that it introduces several unwanted consequences, including:
> - Confusion: For instance, when an LLM enumerates multiple answers, users are unclear which one is correct.
> - Inefficiency: It costs useless tokens by repeating the question that is not necessary.
> - Performance Discrepancy: It shows a significantly different performance than the concise answers, and in most cases is lower than the concise/decisive answers.
>
>
> Therefore, we believe this behavior requires mitigation. Please note that our goal is not to ensure the model is always fully correct but to emphasize that LLMs should respond without verbosity compensation or hesitation, even when errors occur. To sum up, the research question we aim to address is the issue of _harmful indirect answers resulting from verbosity compensation behavior_.
>
> Additionally, the research question of VC behavior is meaningful. In this paper, we define VC, an understuied type of harmful behavior of LLMs, and contribute to bridging the performance, uncertainty, and verbosity compensation behavior together.  These insights can inspire the development of practical applications and effective mitigation strategies.
>
> Applications: VC behavior can serve as an uncertainty evaluation metric. We can apply it in the routing algorithm and surpass previous routing baselines (Appendix C.1 in newer pdf). Furthermore, users querying GPT can require the model to generate as concisely as possible. If the result exceeds the expected length, users may reject the answer or request further refinement. Mitigation: since uncertainty and VC are connected, future studies can improve the uncertainty of the LLMs by improving VC behavior, such as contrastive finetuning on VC samples to make the model more decisive.
>
> Please refer to the introduction in the newer PDF for more details.

---

> ### Author Response · Authors · 2024-11-27
> **Author Response (Part 2)**
>
> ---
>
> > ### **Weakness 2**
> > **The dataset used for this experiment is creates using existing datasets from previous works, most of these are from previous years and there is high chance that these were in the training data that was used for training different LLMs used in the paper and the details of these data-leakage are not explored or mentioned.**
>
> We acknowledge that there might be data leakage. However, data leakage is not a problem for studying our research question because we are not trying to compare the absolute performance of each dataset. We only need to compare the difference between concise and verbose responses on the same dataset. As long as the potential dataset leaked is over the entire dataset, the comparison inside one dataset is still fair.
>
> To further test the influence of data leakage, we evaluate the VC behavior of a newly proposed trip planning dataset from Natural-Plan benchmark [1]. The knowledge cutoff of llama-3 (April 2024) is earlier than the proposal of the dataset (June 2024). So we can safely assume that there is no data leakage in the data. We follow their zero-shot setting and run it on two Llama-3 models and evaluate VC frequency and performance gaps. As shown in the table, VC also happens frequently in trip planning, demonstrating the pervasiveness of VC without data leakage.
>
> |  | concise | verbose | $\Delta$ | Avg. Recall | VC Freq. |
> |---|---|---|---|---|---|
> | llama-3-8b | 15.18 | 3.62 | 11.56 | 9.22 | **51.49** |
> | llama-3-70b | **21.81** | **4.87** | **16.94** | **19.63** | 12.87 |
>
> ---
>
> > ### **Weakness 3**
> > **The prompt used for the experiment is not clear, more details on why 3 words was choose and why is it mentioned multiple times single phrase if possible, but then you mention single phrase and also 3 words. Given that the work depends a lot on the prompt, more prompts should have been experimented and more details of those results are needed.**
>
> **“Why 3 words were chosen”**: As indicated at the beginning of the response, our focus is on harmful indirect responses of verbosity compensation (VC) behaviors, rather than analyzing issues related solely to response length. To solve this, as long as we can observe significant VC,  such as repeating the question, a simpler setting is better because it is clear and can avoid the influence of other factors.
>
> Motivated by this, we choose short-form QA and restrict the QA pairs to concise formats (less than 3 words) so that the verbosity detector is easy to implement and achieves high accuracy. This simpler and controlled dataset setting allows us to better analyze LLM behavior while minimizing the influence of verbosity detector inaccuracies.
>
> Besides, our experiments are not restricted to short-form answers. To demonstrate the generalization of the proposed VC behavior, in the paper, we run the experiments on Chain-of-Though settings where the responses can contain more than 100 words. This table shows the statistics of Chain-of-Thought experiments. The average response length can reach more than 300 words, and the VC behavior is still pervasive.
>
>
> |               |   MMLU  |         |         |         |  Qasper |         |         |         |
> |---------------|:-------:|:-------:|:-------:|:-------:|:-------:|:-------:|:-------:|:-------:|
> |               | VC Freq. | Min Len | Max Len | Avg Len | VC Freq. | Min Len | Max Len | Avg Len |
> | gpt-3.5-turbo |  **51.49** |       3 |      90 |   26.24 |  37.62 |       4 |      81 |   23.38 |
> | gemma-2-9b |  20.79 |       9 |     107 |   27.92 |  43.56 |      18 |     103 |   37.08 |
> | llama-3-8b    |  43.56 |      **15** |     **333** |   **57.14** | **44.15**  |     **20**    |  **185**     |  **50.15**       |
>
> Finally, we also observe VC in the experiment on Natural-Plan in the response to **Weakness2**.

---

> ### Author Response · Authors · 2024-11-27
> **Author Response (Part 3)**
>
> **“Why mentioned multiple times”**: We design a reinforced prompt to ensure LLM understands concise responses are required. Thus, we reinforce the prompt by repetition [1], explanation, etc., especially for the weaker models, making a fairer comparison by avoiding failing to understand instructions.
>
> We further experiment with multiple possible prompts on llama3-8b on the Qasper dataset to show VC is not overfitting to certain prompt settings. We aim to show that as long as the model knows to generate as concise as possible, we can observe significant VC behaviors.
>
> Prompt_2:
> ``` Using a single phrase rather than a sentence. Please answer in 3 words. ```
>
> Prompt_3:
> ```Answer as concisely as possible. ```
>
> The table shows the performance gap on MMLU and Qasper datasets using Llama-3-8b. As can be seen, compared with the original prompt, the variation of the prompt can also observe a significant $\Delta$ over both datasets. This demonstrates the robustness of VC against the choice of prompt.
>
> |  | MMLU |  |  |  | Qasper |  |  |  |
> |---|:---:|:---:|:---:|:---:|:---:|:---:|:---:|:---:|
> |  | Concise | Verbose | $\Delta$ | Avg. | Concise | Verbose | $\Delta$ | Avg. |
> | orginal | 58.4 | 44.82 | 13.57 | 55.6 | 58.99 | 54.6 | 4.39 | 55.98 |
> | prompt_2 | 55.13 | 43.43 | 11.71 | 52.70 | 54.22 | 48.11 | 6.11 | 51.30 |
> | prompt_3 | **68.04** | **50.26** | **17.78** | **61.07** | **70.17** | **60.44** | **9.73** | **63.63** |
>
>
> ---
>
> > ### **Weakness 4**
> > **Though there is mention of verbosity detector the actual definition and formula on how you calculate it is missing.**
>
> In our experiments, we design simple reference-free metrics. For short-form QA settings, it is $|r| >3$, meaning more than 3 tokens in the response $r$. For CoT settings, it is $\left(\bigvee_{s\in S} |s| > 10\right)$, where we use $(*)$ to output 1 if * is true, otherwise 0, showing that each step should have less than 10 tokens. We will clarify it in the new version.
>
> ---
>
> > ### **Weakness 5**
> >  **There is introduction of algorithm, the paper mentions that VC might causes latency, there can be latency due to algorithm as well.**
>
> We claim that the time wasted on generating _useless (non-informative)_ tokens is greatly reduced by the proposed algorithm compared with individual models. With our proposed cascade algorithm, the total inference time might be higher than using a small model due to the use of the large model (0.79 -> 1.21 per sample), but the number of useless tokens generated is much less. To evaluate this, we further assess the number of useless tokens generated. Given a response $r$, we first define the useless tokens as the part with longer than 3 tokens in response $r$:
> $$ \text{total number of useless tokens} = \sum_{i=1}^N  \max(0, |r_i| - 3), $$
> where $N$ is the number of samples in a dataset.
>
> Then, we evaluate the total useless tokens generated by individual models and proposed cascade algorithms. As shown in the table, by using the proposed algorithm, the useless tokens generated decrease from 596/327 to 93, mitigating the VC rate from 41.4% to 21.0% on the NarrativeQA dataset, demonstrating that useless tokens greatly decrease by using the proposed algorithm.
>
> |  | Qasper |  |  |  | NarrativeQA |  |  |  |
> |---|---|---|---|---|---|---|---|---|
> |  | Mistral useless | GPT useless | Total | VC Freq. | Mistral useless| GPT useless| Total | VC Freq. | Time per sample (sec) |
> | mistral | 663 | 0 | 663 | 63.81 | 596 | 0 | 596 | 41.4 | **0.79** |
> | gpt-4o | 0 | 207 | 207 | 31.79 | 0 | 327 | 327 | 50.4 | 1.27 |
> | mistral->gpt-4o | 0 | 86 | **86** | **16.6** | 0 | 93 | **93** | **21.0** | 1.21 |
>
> Additionally, this algorithm improves other aspects. It can greatly improve performance because it can work as a routing algorithm (Section 5.4 in the paper). This also alleviates the cognitive load of the users which demonstrates the helpfulness to the users (examples at the beginning of the response).
>
> ---
>
> > ### **Weakness 6**
> >  **Writing can be improved**
>
>
> We will be more than happy to clarify any points! We have rewritten the abstract as well as the introduction by clearly stating the research question and its importance. We also submit the revised version in the pdf.

---

> ### Author Response · Authors · 2024-11-27
> **Author Response (Part 4)**
>
> ---
>
> > ### **Weakness 7**
> >  **Most of the examples mentioned in the paper have some numbers in the answers, out of 3 correctly given answers 2 doesn't have any numbers, given that there is existing research that LLMs are not good at the numbers, it would be better to compare the behavior on those lines as well, if the output is wrong because of the numbers and not other factors**
>
> This is an interesting analysis. We further analyze the performance and VC frequency of the samples with and without numbers using llama-3-8b on Qasper dataset. The results are shown in the table, we find that although the model is easier to perform better on the sample without numbers, the VC frequency is relatively lower for the responses with digits. To understand the reason, we further inspect the Qasper dataset, we find that the samples with numbers are not as open-ended as the ones without numbers, meaning that the search space of the answers with numbers is smaller. This leads to a lower VC frequency and is easier to answer.
>
> |  | concise | verbose | VC Freq. | recall |
> |---|---|---|---|---|
> | **Qasper** |  |  |  |  |
> | w/o digits | **58.99** | 53.66| 52.63 | **56.18** |
> | w/ digits | 58.97 | **57.73** | **45.83** | 58.40 |
> | **NarrativeQA** |  |  |  |  |
> | w/o digits | **33.39** | 18.18 | 40.66 | **27.21** |
> | w/ digits | 56.25 | **10.00** | **38.46** | 38.46 |
>
>
> [1] Zheng H S, Mishra S, Zhang H, et al. NATURAL PLAN: Benchmarking LLMs on Natural Language Planning[J]. arXiv preprint arXiv:2406.04520, 2024.
>
> [2] Agrawal D, Gao S, Gajek M. Can't Remember Details in Long Documents? You Need Some R&R[J]. arXiv preprint arXiv:2403.05004, 2024.

---

> > ### Comment · Reviewer_esHV · 2024-11-30
> > **Increased the score to 5**
> >
> > Thank you for the considerations and explanations,  I have increased my score to 5.

---

> > > ### Author Response · Authors · 2024-12-04
> > > **Thanks from Authors**
> > >
> > > We are glad to know our explanations help address your concerns! We would also like to thank you for your thoughtful questions and insightful discussion that helped us to shape a clearer and stronger paper.

---

### Official Review · Reviewer_Tbb5 · 2024-11-03

**Soundness:** 3
**Presentation:** 3
**Contribution:** 3
**Rating:** 8
**Confidence:** 4

**Summary:**

This paper studied the verbosity issue of LLMs. The authors tested the frequency that LLMs generate verbose answers (a behavior defined as **verbosity compensation** by the authors), and showed the following findings:

(1) All mainstream LLMs generate verbose answers.
(2) Models have different performance when generating concise and verbose answers. Most models perform worse when their response are verbose.
(3) There is a positive correlation between answer verbosity and model uncertainty. This means that models are usually more uncertain about the answers when they generate excessively long answers.
(4) Using verbosity as an indicator to route models (if the smaller model generates a verbose answer, then we give this example to the larger model) is a promising approach to balance model performance and inference cost.

**Strengths:**

1. The authors are the first to formally define the verbosity compensation (VC) behavior of LLMs which refers to generating an answer that is more verbose than required.
2. The authors collected a benchmark from existing NLP task data to analyze the frequency of VC in LLMs. They further designed metrics to calculate the impact of VC on model performance.
3. The authors did extensive experiments to test the correlation between VC and (1) model performance (2) model uncertainty. These experiments led to the findings I listed in paper summary.
4. The authors proposed a simple but effective model routing technique based on their findings on VC: routing the instance to stronger models if the weaker model exhibits verbosity in answering. They further tested its effectiveness in reducing verbosity and balacing performance-cost trade-off.
5. The paper is well written and the experimental results are clearly presented.

**Weaknesses:**

1. The fairness of evaluation metrics is questionable. This includes two aspects:

(1) The authors wrote in section 3.1: *We define a response $r$ to be verbose if and only if it can be further compressed with fewer tokens
while keeping the same meaning.* However, in actual experiments, they directly used "the number of tokens in the response" as the verbosity detector. **They did not measure "whether the answer can be compressed while keeping the same meaning"**. This may contain bias, for example, if the gold answer is 3 tokens but the model generates a wrong answer that is 5 tokens, although the answer is wrong, but I don't think the wrong answer is verbose if it cannot be further simplified. However, such a case will be identified as verbose by the authors as their criteria of conciseness is "less than 4 words".

(2) In section 3.2, the authors define $\Delta$ (performance difference) as the model score on cases where its answer is verbose minus the score on cases where its answer is concise. This means, **the two scores are not calculated on the same test instances**. First, I suspect this is not 100% fair, because there may be only a few "verbose instances" or "concise instances" for some LLMs, which will lead to inaccurate estimation. Second, I suspect this metric can be affected by spurious correlations. The authors showed in section 5.3 that generating verbose answers indicate a model's uncertainty. And it is possible that model-uncertain cases are those that are more difficult. Therefore, it is reasonable that models perform worse on such instances.

2. The evaluation set is limited to scenarios where the gold answer is less than 4 words. However, I think such short-form QA is just one of the possible situations where verbosity may occur, and it might not even be the most critical one. Verbosity may be more harmful in open-ended generation cases that are not evaluated in this paper. For example, *repeating the question when answering a factual question* may be less annoying than *saying a lot of useless words when planning a trip*.

3. When evaluating the proposed cascading approach in model routing, the authors only compared it with randomly selecting models. However, they did not compare with more related baselines such as "model routing based on response uncertainty". For example, if the weaker model is uncertain about the response, then give this question to the stronger model.

**Questions:**

1. The equation in section 3.2 uses the score of cases where $V(x,y,r)=1$ as the minuend, and the score of cases where $V(x,y,r)=0$ as the subtrahend. However, the authors defined the formula as "the average score of the concise responses minus the average score of the verbose responses". Should the minuend and subtrahend be reversed?
2. I have concerns about the definition of verbosity. From Figure 1 and the definition of verbosity detector, it seems like answers like *Paris is the capital of France* will be judged as verbose for the question *Which city is the capital of France?*. However, I think such a response seems pretty natural and human-like, and it does not bring any confusion or inefficiency to the communication. Therefore, I wonder whether the definition of "if the gold answer is less than 4 words, then any answer longer than 4 words is verbose" is too strict and whether it will be widely accepted.

---

> ### Author Response · Authors · 2024-11-27
> **Author Response (Part 1)**
>
> We thank the reviewer for the helpful and constructive feedback! We will state the research problem more clearly and answer the questions one by one in the response. We also submitted another pdf that addresses your comments.
>
> ---
>
> First, we would state our research problem and metric design more clearly.
>
> The conventional understanding of verbosity primarily focuses on the **lengthiness** of responses. In contrast, our work highlights the **undesired indirect or hesitant behavior** of Verbosity Compensation exhibited by LLMs. This behavior goes beyond mere length and includes tendencies such as repeating questions, introducing ambiguity, or providing excessive enumeration when tasked with generating concise answers.  While we use response **length** as a simple yet effective **tool** to identify such behavior, the core contribution of our work is independent of length. To clearly differentiate from the lengthy issue, our proposed Verbosity Compensation specifically emphasizes an *undesired behavior of not giving a valid answer to the question directly and decisively*.
>
>
> To solve this, as long as we can observe significant VC,  such as repeating the question, a simpler setting is better because it is clear and can avoid the influence of other factors. Motivated by this, we choose short-form QA and restrict the QA pairs to concise formats (less than 3 words) so that the verbosity detector is easy to implement and achieves high accuracy because the human study demonstrates that in this case, longer than 3 tokens can work as a good indicator of VC behavior. This design avoids the influence of the inaccuracy of the verbosity detector.
>
> ---
>
> Next, we would answer the questions one by one:
>
>
> > ### **Weakness 1**
> > **They did not measure "whether the answer can be compressed while keeping the same meaning".**
>
> The measure of "whether the answer can be compressed while retaining the same meaning" is conducted in a controlled setting across both short-form QA and Chain-of-Thought experiments. We acknowledge that without such control, this measure cannot be naturally assessed.
>
> For short-form QA, we restrict the QA pairs to concise formats, intentionally selecting samples where responses exceeding three words are verbose and can be compressed without any loss of information. As outlined earlier, this simpler and controlled dataset setting allows us to better analyze LLM behavior while minimizing the influence of verbosity detector inaccuracies.
>
> For the Chain-of-Thought setting, we limit each step to fewer than 10 words but do not restrict the total number of steps. In this case, if the model generates a step exceeding 10 words, it is ensured that the step can be compressed by assigning part of the response to the next step.
>
>
> ---
>
> > ### **Weakness 1 (1)**
> > **the two scores are not calculated on the same test instances**
>
> To prevent the influence of different test instances, we further conduct an analysis that fixes the data point and compares different models. Specifically, for each data point, we calculated the ratio of LLMs exhibiting VC behavior and reported the averaged ratio across datasets in the table. This approach also increases the robustness of our findings, as the support (number of samples) for each dataset is 14 times higher than when using a single model. As shown in the table, for all 5 datasets, the performance $\Delta$ is still significant.
>
> |             | Concise Rec. | # Concise  | Verbose Rec. | # Verbose | $\Delta$ | VC Freq. | Avg. Recall |
> |-------------|----------------:|-----------------:|----------------:|-----------------:|--------:|---------:|-------------:|
> | MMLU        |         65.09 |            1694  |         51.47 |         475    | 13.62 |  24.20 |      61.79 |
> | NQ30           |         53.34 |          1516    |         44.89 |        362     |  8.45 |  26.41 |      51.10 |
> | NarrativeQA |         38.09 |           2540   |         31.67 |        355     |  6.42 |  **36.29** |      35.76 |
> | Qasper        |         61.85 |            2272  |         45.63 |         389    | **16.22** |  32.46 |      56.59 |
> | LongBench   |         50.31 |          1912    |         44.22 |         375    |  6.10 |  30.42 |      48.46 |

---

> ### Author Response · Authors · 2024-11-27
> **Author Response (Part 2)**
>
> > ### **Weakness 1(2)**
> > **there may be only a few "verbose instances" or "concise instances" for some LLMs.**
>
> We agree in a normal setting, some samples are easier to answer verbosely. However, there are two aspects in our setting to prevent this situation. First, we filter out the sample with no more than 3 words, meaning that all samples can be answered with a concise answer. Second, we claim that all samples can be answered concisely. As shown in the example:
>
> What is the capital city of the United States? Answer as concisely as possible:
>
> Some acceptable answers:
> - Washington DC (Correct and concise)
> - Not sure (Confessing limit and concise)
> - Paris (Wrong but concise)
>
> As outlined, we are not assuming the model can be fully correct, but it should be capable of answering it wrong but concisely or confessing its limits concisely.
>
> ---
>
> > ### **Weakness 1(2)**
> > **Second, I suspect this metric can be affected by spurious correlations. The authors showed in section 5.3 that generating verbose answers indicates a model's uncertainty. And it is possible that model-uncertain cases are those that are more difficult. Therefore, it is reasonable that models perform worse on such instances.**
>
> The correlation between difficulties and uncertainty is part of what we want to analyze. We connect the verbosity behavior with performance (difficulties) and uncertainty to show their relationship.  In fact, we intentionally set the different difficulty levels in the dataset so that we can see the contrast in performance. Otherwise, all samples are perfectly correct/wrong and VC is not observable. We agree that uncertainty and model performance are correlated, but we incorporate verbosity compensation behavior to connect three factors that are not shown in previous work.
>
> ---
>
> > ### **Weakness 2**
> > **The evaluation set is limited to scenarios where the gold answer is less than 4 words. However, I think such short-form QA is just one of the possible situations where verbosity may occur, and it might not even be the most critical one. Verbosity may be more harmful in open-ended generation cases that are not evaluated in this paper. For example, repeating the question when answering a factual question may be less annoying than saying a lot of useless words when planning a trip.**
>
> We fully agree with you that verbosity compensation behavior is not limited to short-form QA. Our evaluation is not limited to gold answers with less than 4 words. To demonstrate the generalization of the proposed VC behavior, in the paper, we run the experiments on Chain-of-Though settings where the responses can contain more than 100 words. This table shows the statistics of Chain-of-Thought experiments. The average response length can reach more than 50 words, and the VC behavior is still pervasive.
>
> |               |   MMLU  |         |         |         |  Qasper |         |         |         |
> |---------------|:-------:|:-------:|:-------:|:-------:|:-------:|:-------:|:-------:|:-------:|
> |               | VC Freq | Min Len | Max Len | Avg Len | VC Freq | Min Len | Max Len | Avg Len |
> | gpt-3.5-turbo |  **51.49** |       3 |      90 |   26.24 |  37.62 |       4 |      81 |   23.38 |
> | gemma-2-9b |  20.79 |       9 |     107 |   27.92 |  43.56 |      18 |     103 |   37.08 |
> | llama-3-8b    |  43.56 |      **15** |     **333** |   **57.14** | **44.15**  |     **20**    |  **185**     |  **50.15**       |
>
> However,  as stated at the beginning, we want to show the behavior, thus, a simpler dataset might be better for analysis and avoid the influence of external factors, such as differences in oracle length in the results.
>
> We agree the open-ended domains are an interesting direction! Following your suggestion, we further evaluate the VC behavior on the trip planning dataset from the Natural-Plan benchmark [1] using two Llama-3 models and test VC frequency and performance gaps. As shown in the table, the VC is still pervasive, reaching 16.94% for the Llama-3-70b model.
>
> |  | concise | verbose | $\Delta$ | Avg. Recall | VC Freq. |
> |---|---|---|---|---|---|
> | llama-3-8b | 15.18 | 3.62 | 11.56 | 9.22 | **51.49** |
> | llama-3-70b | **21.81** | **4.87** | **16.94** | **19.63** | 12.87 |

---

> ### Author Response · Authors · 2024-11-27
> **Author Response (Part 3)**
>
> > ### **Weakness 3**
> > **When evaluating the proposed cascading approach in model routing, the authors only compared it with randomly selecting models. However, they did not compare with more related baselines such as "model routing based on response uncertainty". For example, if the weaker model is uncertain about the response, then give this question to the stronger model.**
>
> Thanks for your suggestion! We further conduct an analysis to compare the performance of the uncertainty-based routing algorithm as well as the random baselines. The results are shown in Appendix C.1 in the newer pdf. As can be seen in the Figure 7, although the uncertainty-based routing algorithm can obtain a better performance than the random baseline, the proposed algorithm still outperforms this approach by comparing the AUC of the figure (Area Under the Curve), demonstrating the effectiveness of the proposed algorithm.
>
> ---
>
> > **Q1:  Should the minuend and subtrahend be reversed?**
>
> Thanks for bringing this issue up! We reverse the minuend and subtrahend in equation 3.2 and modify this in the pdf.
>
> ---
>
> > **Q2:  I wonder whether the definition of "if the gold answer is less than 4 words, then any answer longer than 4 words is verbose" is too strict and whether it will be widely accepted.**
>
> As shown in Table 7 in the appendix, we ask the model to answer as concisely as possible and not to repeat the questions. Under this constraint, the response by repeating the question is not natural because it does not follow the instructions. As shown in the CoT experiment of weakness 2, we use different settings for CoT experiments that can contain more than 100 words in the responses.
>
> [1] Zheng H S, Mishra S, Zhang H, et al. NATURAL PLAN: Benchmarking LLMs on Natural Language Planning[J]. arXiv preprint arXiv:2406.04520, 2024.

---

> > ### Comment · Reviewer_Tbb5 · 2024-11-30
> > **Response to Authors**
> >
> > I appreciate the authors' response to my review.
> >
> > First, the additional experiments shown by the authors, (1) model performance comparison under the same test instances (2) VC analysis on open-ended generation tasks (3) VC-based routing vs. uncertainty-based routing, addressed my corresponding concerns in weakness 1(2), weakness 2 and weakness 3.
> >
> > Besides, the authors also responded to my concerns about the VC criteria. Frankly speaking, I think their statement that "we carefully selected examples so that the models can express the correct answer within 3 words. All answers > 3 words can be losslessly compressed and the models will not generate wrong answers > 3 words" is just too assertive to justify that such a criteria covers diverse scenarios. And the practice of "intentionally sample instances" is hard to convince every reader. After all, one can only evaluate a limited number of models in such a study, and the selected data will only cover a small portion of real-world application scenarios, so it is hard to say "such a criterion works without many exceptions on our tested models and selected data, so it is a valuable finding that mirrors all other models and all real-world scenarios".
> >
> > The authors mentioned that there is a human study about whether 3 words is a good criteria, but I could not find the relevant information in the paper. What human study did the authors carry out?

---

> > > ### Author Response · Authors · 2024-12-02
> > > **Author Response (Part 2)**
> > >
> > > VC criteria: Any step contains more than k words. k is the maximum length of the ground truth steps.
> > >
> > > > ### Question 3:
> > > > **What human study did the authors carry out?**
> > >
> > > For the human study on short-form QA, we select 30 samples from each dataset and inspect the quality of the samples. Specifically,  human annotators judge "whether any valid answer to this question can be described in 3 tokens (or any answer that is longer than 3 tokens can be compressed without information loss)".
> > >
> > > Here are some examples of ground truth answers within 3 words, which can be entities, numbers, or dates:
> > >
> > > ```
> > > Sample 1:
> > > Q: Which language has the lowest error rate reduction?
> > > GT: Thai
> > >
> > > Sample 2:
> > > Q: When did William Le Poer Trench's father die?
> > > GT: 26 April 1872
> > >
> > > Sample 3:
> > > Q: What is the greatest possible quotient of any two distinct members of the set $\{\frac{2}{5}, \frac{1}{2},5,10\}$? Specifically, we wish to maximize $\frac{x}{y}$, where $x$ and $y$ are chosen from the previous set.
> > > GT: 25
> > > ```

---

> > > > ### Comment · Reviewer_Tbb5 · 2024-12-03
> > > > **Response to Authors**
> > > >
> > > > I appreciate the authors for the additional examples and experimental results. They made the paper more comprehensive and solid.
> > > >
> > > > I still have a question about the following example shown in your response:
> > > > ```
> > > > Q: What language(s) is/are represented in the dataset?
> > > > GT: English
> > > > Pred: English , German , French , Greek .
> > > > ```
> > > > Based on my understanding, the prediction is wrong because incorrect languages are predicted. But I doubt if this answer is *verbose* because I don't think it can be further compressed without losing any of its meaning. It looks like there is not a proper method to judge whether such long answers is verbose. According to the submission paper:
> > > > ```
> > > > VC emphasizes the detailed behavior of generating compressible tokens with a low density of useful information. (line 52)
> > > > ```
> > > > How should we distinguish "compressible answer with not much useful information" and "wrong answer with incorrect information"?

---

> > > > > ### Author Response · Authors · 2024-12-03
> > > > > **Author Response**
> > > > >
> > > > > We are delighted that our additional examples and experimental results strengthen the paper!
> > > > >
> > > > > > **How should we distinguish "compressible answer with not much useful information" and "wrong answer with incorrect information"?**
> > > > >
> > > > > As we stated, VC criteria for Short-form QA is:
> > > > >
> > > > > verbose if the response is longer than the maximum length of the ground truth (3 words in our original submission).
> > > > >
> > > > > Under our VC criteria, we can distinguish between "compressible answer with not much useful information" and "wrong answer with incorrect information" as shown in the following examples:
> > > > >
> > > > > ```
> > > > > Q: What language(s) is/are represented in the dataset?
> > > > > GT: English
> > > > > Pred: English is represented in the dataset. -> Verbose and correct
> > > > > Pred: Chinese, German , French , Greek . -> Verbose and incorrect
> > > > > Pred: English. -> Not verbose and correct
> > > > > Pred: German. -> Not verbose and incorrect
> > > > > ```
> > > > > We note that verbosity and incorrectness are separate dimensions, and the contribution of our paper is to demonstrate that they are strongly correlated in LLM responses.
> > > > >
> > > > > > **But I doubt if this answer is verbose**
> > > > >
> > > > > We classify this sample as the verbosity (enumerating) because of listing more languages **than ground truth**. We will modify the paper so that VC is more clearly defined: "generating compressible tokens with a lower density of useful information than ground truth (assuming ground truth is fully compressed)".

---

> > > > > > ### Comment · Reviewer_Tbb5 · 2024-12-03
> > > > > > **Response to Authors**
> > > > > >
> > > > > > Thank you for your clarifications. I will raise my score to 8 given all the responses, clarifications, and experiments that the authors provided in the discussion phase.

---

> > > > > > > ### Author Response · Authors · 2024-12-04
> > > > > > > **Thanks from Authors**
> > > > > > >
> > > > > > > We are really glad that our responses address your concerns! Thanks so much for your thoughtful questions and insightful discussion. We will carefully include our new discussion in the final version. We deeply thank the effort you made during the whole process!

---

> ### Author Response · Authors · 2024-12-02
> **Author Response (Part 1)**
>
> Thanks for the insightful comments and questions. Your valuable feedback greatly enhances our work! We are glad that our additional experiments addressed your concerns in weakness 1(2), weakness 2, and weakness 3.
>
> For your concern on VC criteria in weakness 1(1), we would like to clarify that we used different VC criteria in our experiments as summarized here:
>
> - **Short-form QA**: verbose if the response is longer than the maximum length of the ground truth length (3 words in our original submission; more experiments below on longer answers per your request);
> - **CoT Reasoning**: verbose if any step contains more than k words, where k=10 indicates the number of tokens in the longest reasoning step in the ground truth;
> - **Planning**: the same as CoT Reasoning.
>
> We believe that these VC criteria can appropriately assess the verbosity of LLM responses under diverse tasks. Next, we response the detailed questions:
>
> > ### Question 1:
> > **the practice of "intentionally sample instances" is hard to convince every reader**
>
> For short-form QA, to demonstrate the generalization of our VC criteria, we further conduct the experiments on the original dataset _without any sample selection_. Thus, it contains answers with various lengths. Without losing the generality, we set the threshold of verbose and concise responses according to the length of the ground truth answer. And according to the suggestion of Reviewer esHV, we remove the prompt “answer in 3 words” and only keep the prompt “Answer as concisely as possible”. The table below shows the results. As can be seen, for the samples where the ground truth contains more than 3 words, the VC is consistent with the short-form QA, and is even more significant. Thus, we use short samples as representative data to observe the VC behavior while simplifying the evaluation. We will add the evaluation of the entire datasets in the final version.
>
> |  | Qasper |  |  |  | LongBench |  |  |  |
> |---|---|---|---|---|---|---|---|---|
> |  | concise | verbose | delta | average | concise | verbose | delta | average |
> | gt < 3 words | 58.23 | 56.02 | 2.20 | 55.98 | 44.32 | 33.14 | 11.18 | 37.75 |
> | gt > 3 words | 45.80 | 33.30 | **12.50** | 43.52 | 53.12 | 39.45 | **13.67** | 51.25 |
> | Average | 51.71 | 46.98 | 4.73 | 48.71 | 47.16 | 33.55 | 13.61 | 40.24 |
>
> Besides, we also want to clarify that our evaluated settings on CoT reasoning and trip planning did not filter out samples.
>
>
> > ### Question 2:
> > **just too assertive to justify that such a criteria covers diverse scenarios**
>
> We agree that 3-word criteria cannot cover all datasets with various lengths. However, we only claim the 3-word criteria is reasonable on the short-form QA. Our human study and samples show the reasonability of such a choice (see Question 3). For the rest of the scenarios, we utilize other evaluation metrics in other datasets with longer responses. Here is a list of VC criteria and types of examples we have (Q: Question, GT: Ground Truth, Pred: Prediction):
>
> - Short-form QA with ground truth < 3 words:
> ```
> Q: What language(s) is/are represented in the dataset?
> GT: English
> Pred: English , German , French , Greek .
> ```
>
> VC criteria (used in the paper): containing more than 3 words
>
> - Short-form QA with ground truth > 3 words:
> ```
> Q: What controversial topics are experimented with?
> GT: political events such as elections , corruption cases or justice decisions
> Pred: The controversial topics experimented with include the Israeli-Palestinian conflict , the 2016 US presidential election , the Bolsonaro presidential campaign , the Ukraine-Russia conflict , and the Netanyahu corruption case .
> ```
> VC criteria: containing more tokens than ground truth:
>
>
>
> - Chain-of-Thought:
> ```
> Q: An astronaut lands on a planet whose mass and radius are each twice that of Earth. If the astronaut weighs 800 N on Earth, how much will he weigh on this planet?
> GT: 400 N
> Pred:
> Here's the solution:
>
> 1. Astronaut's weight on Earth is 800 N.
> 2. Planet's mass is twice Earth's, so astronaut's weight on planet is 2 × 800 N = 1600 N.
> 3. Answer: 1600 N.
> ```
> VC criteria (used in the paper):  Any step contains more than k words. k is the maximum length of the ground truth steps. Given that MMLU does not provide reasoning steps as ground truth, we assume that the steps can always be decomposed into sub-steps with less than k=10 words, and ask the model to generate each step in less than 10 words.
>
> - Planning:
> ```
> Q: Find a trip plan of visiting the cities for 25 days by taking direct flights to commute between them.
> GT:
> Day 1-2: Arriving in Barcelona for 2 days.
> Day 2: Fly from Barcelona to Prague.
> Day 2-5: Visit Prague for 4 days.
> Day 5: Fly from Prague to Edinburgh.
> …
> Pred:
> Step 1: Start in Bucharest for 5 days.
> Step 2: Take a direct flight to Lyon for 4 days.
> Step 3: Take a direct flight to Barcelona for 2 days, attending the workshop.
> …
> ```

---

### Official Review · Reviewer_3ZZU · 2024-11-04

**Soundness:** 2
**Presentation:** 3
**Contribution:** 2
**Rating:** 5
**Confidence:** 3

**Summary:**

This paper proposes a concept termed “Verbosity Compensation” (VC), where LLMs respond with overly verbose answers to compensate for uncertainty. VC is shown to decrease response efficiency, increase costs, and obscure clarity, highlighting an urgent need to mitigate this behavior. To address VC, the paper introduces a cascade algorithm that effectively reduces verbosity by replacing verbose responses with alternative model-generated responses, achieving a significant reduction in VC, particularly demonstrated with a 63.81% to 16.16% decrease for the Mistral model on the Qasper dataset.

**Strengths:**

Interesting research problem.

**Weaknesses:**

1. In the verbosity compensation and uncertainty section, how to quantity the uncertainty for each split is not clear. How to get the perplexity evaluation metric and eigenvalues is not clear.
2. In my understanding, In performance difference calculation, the recall is calculated by exact matching, checking if the generation contains the exact same string of the ground truth answer. There might be some correct but not exactly matched answer in LLM generation, especially in the lengthy (verbose) response. So would be better to design a partial credit to further validate the findings of this work.
3. The proposed cascade model selection method can leak the ground truth answer. In the selection process, the detector would take ground truth answer as input for answer selection.
4. In section 4.2, what does the |r|>3 mean?
5. In line 365 and figure 3, what does the llama-3-80b mean? I believe llama-3 doesn’t have the 80B version.

**Questions:**

please see the weaknesses above

---

> ### Author Response · Authors · 2024-11-27
> **Author Response (Part 1)**
>
> We thank the reviewer for the helpful feedback! We will address each of your concerns in the response one by one. We also submitted another pdf that was revised according to your valuable comments.
>
> ---
>
> > Interesting research problem.
>
> Thanks for finding our research interesting! We would emphasize that Verbosity Compensation is an important and useful topic as well. In this paper, we define Verbosity Compensation (VC), an understudied type of harmful behavior of LLMs, and contribute to bridging the performance, uncertainty, and VC together.  These insights can inspire the development of practical applications and effective mitigation strategies.
>
> **Applications**: VC behavior can serve as an uncertainty evaluation metric. We can apply it in the routing algorithm and surpass previous routing baselines (Appendix C.1 in newer pdf). Furthermore, users querying GPT can require the model to generate as concisely as possible. If the result exceeds the expected length, users may reject the answer or request further refinement.
>
> **Mitigation**: since uncertainty and VC are connected, we can mitigate the uncertainty of the LLMs by alleviating VC behavior, such as contrastive finetuning on VC samples to make the model more decisive.
>
> ---
>
> > ### **Weakness 1**
> > **In the verbosity compensation and uncertainty section, how to quantity the uncertainty for each split is not clear. How to get the perplexity evaluation metric and eigenvalues is not clear.**
>
> We quantify the uncertainty by computing the averaged uncertainty score over the sample from all datasets (around 2000 data points in total). Specifically, for each model, we obtain the responses for all 5 datasets, and group the responses according to the number of its token. Each group contains responses of the same length. Then, we apply evaluation metrics on each response to obtain the uncertainty score and average it across all responses in the group to get the final score. We use two types of uncertainty scores for open/closed-sourced models:
>
> For perplexity, we compute the probability of generating the sequence by softmax distribution provided by white-box models, denoted as:
> $$ \text{Perplexity} = \text{exp}(-\frac{1}{N} \sum_{i=1}^N \log p(w_i)), $$
>
> Where $w_i$ is the $i$-th token in response and $N$ is the total number of the tokens.
>
> For eigenvalues, following [1], we ask the LLM to generate the results 10 times to obtain $r_1,r_2,...,r_{10}$ and calculate the diversity of responses by counting the number of its spectral clusters. Specifically, we use NLI model (following their paper, we use roberta-large-mnli) to compute a 10x10 matrix $W$, where each element in $W$ is $(NLI(r_i,r_j) + NLI(r_j,r_i))/2$, showing the entailment of each pair of the response. Then, this matrix is viewed as an adjacent matrix with 10 vertexes in the graph of entailment relations, we can apply spectral clustering over it. According to the theorem [2], we compute the sum of eigenvalues of the graph laplacian which is equal to the number of clusters we want, showing how diverse the responses are.
>
> ---
>
> > ### **Weakness 2**
> > **In my understanding, In performance difference calculation, the recall is calculated by exact matching, checking if the generation contains the exact same string of the ground truth answer. There might be some correct but not exactly matched answer in LLM generation, especially in the lengthy (verbose) response. So would be better to design a partial credit to further validate the findings of this work.**
>
> The recall we use is actually not the exact matching of strings. It is based on the exact matching of tokens and thus supports partial correctness. For instance, if the gold answer is 15, and the prediction is “The output contains 15 words”. Then the recall of the prediction is 100%. We use recall for evaluating the correlation between VC and performance because it can remove the influence of the output length so that verbose answers won’t get biased.
>
> ---
>
> > ### **Weakness 3**
> > **The proposed cascade model selection method can leak the ground truth answer. In the selection process, the detector would take ground truth answer as input for answer selection.**
>
> We would like to clarify that the ground truth $y$ in the verbosity detector is optional thus the algorithm won’t leak the ground truth if $y$ is not provided. The verbosity detectors can work in a reference-free manner, meaning that the algorithm can work without the gold answer. In our experiments, we use 3 tokens as criteria without using $y$, and the length of the concise response is easy to judge without knowing the groud truth as well because the question usually asks about single entities, like numbers, organizations, and locations. For more settings, we can develop other verbosity detectors that are reference-free. For instance, we can use GPT to detect if the answer can be further compressed to get a score.

---

> ### Author Response · Authors · 2024-11-27
> **Author Response (Part 2)**
>
> > ### **Weakness 4**
> > **In section 4.2, what does the |r|>3 mean?**:
>
> It means the length of the responses is larger than 3. We will clarify this in the later version.
>
> ---
>
> > ### **Weakness 5**
> > **In line 365 and figure 3, what does the llama-3-80b mean? I believe llama-3 doesn’t have the 80B version.**
>
> We will correct this, it is llama-3-70b as shown in the appendix. Thanks!
>
> [1] Lin Z, Trivedi S, Sun J. Generating with confidence: Uncertainty quantification for black-box large language models[J]. arXiv preprint arXiv:2305.19187, 2023.
>
> [2] Ulrike Von Luxburg. A tutorial on spectral clustering. Statistics and computing, 17:395–416, 2007

---

> ### Author Response · Authors · 2024-12-02
> **Kind Reminder from Authors**
>
> Dear Reviewer 3ZZU,
>
> We sincerely appreciate your valuable feedback. We would like to kindly inquire about the extent to which we have successfully addressed the concerns outlined in your review. We greatly value your feedback and would appreciate any further questions or comments you might have.
>
> Thank you again for your time and consideration.
>
> Sincerely,
> All Authors

---

### Author Response · Authors · 2024-12-04
**Summary of the Discussion (Part 1)**

We deeply thank all the valuable feedback and comments from reviewers that have helped to improve our paper! The questions and discussion greatly enhance our work. Throughout the author-reviewer discussion period, we are sincerely glad that reviewers acknowledged our efforts in the responses, clarifications, and experiments that addressed the concerns. We also thank the reviewers for appreciating the interesting concept of Verbosity Compensation (VC) (Reviewer 3ZZU, Tbb5, esHV), extensive benchmarking on 14 models (Reviewer Tbb5, esHV), in-depth analysis and findings on VC (Reviewer Tbb5, esHV), a simple but effective mitigation algorithm (Reviewer Tbb5, esHV) as well as clear presentation and writing (Reviewer Tbb5).

We’d like to summarize our discussion to address the reviewer’s concerns:

> ### **Research problem of the paper [Reviewer Tbb5, esHV]**

The conventional understanding of verbosity primarily focuses on the **lengthiness** of responses. In contrast, our work highlights the **undesired indirect or hesitant responses** of Verbosity Compensation exhibited by LLMs. This behavior goes beyond mere length and includes tendencies such as repeating questions, introducing ambiguity, or providing excessive enumeration when tasked with generating concise answers.  While we use response **length** as a simple yet effective **tool** to identify such behavior, the core contribution of our work focuses on in-depth analysis of behaviors. To clearly differentiate from the lengthy issue, our proposed Verbosity Compensation specifically emphasizes an *undesired behavior of not generating a valid answer to the question directly and decisively*.

> ### **Significance of the research problem [Reviewer 3ZZU, esHV]**

The research question of VC behavior is meaningful. In this paper, we define VC, an understudied type of harmful behavior of LLMs, and contribute to bridging the performance, uncertainty, and verbosity compensation behavior together.  These insights can inspire the development of practical applications and effective mitigation strategies.

Applications: VC behavior can serve as an uncertainty evaluation metric. We can apply it in the routing algorithm and surpass previous routing baselines (Appendix C.1 in newer pdf). Furthermore, users querying GPT can require the model to generate as concisely as possible. If the result exceeds the expected length, users may reject the answer or request further refinement.

Mitigation: since uncertainty and VC are connected, future studies can improve the uncertainty of the LLMs by improving VC behavior, such as contrastive finetuning on VC samples to make the model more decisive.

> ### **Detector of Verbosity Compensation and broadness of tasks [Reviewer Tbb5, esHV]**

We evaluate VC on a broad range of datasets of three settings with diverse response lengths, including Short-form QA, CoT reasoning, and Planning (response length ranging from < 3 words, to around 300 words). For the verbosity detector, we used different VC criteria in our experiments as summarized here:

- **Short-form QA**: verbose if the response is longer than the maximum length of the ground truth length (3 words in our original submission; more experiments below on longer answers per your request);
- **CoT Reasoning**: verbose if any step contains more than k words, where k=10 indicates the number of tokens in the longest reasoning step in the ground truth;
- **Planning**: the same as CoT Reasoning.

We believe that these VC criteria can appropriately assess the verbosity of LLM responses under diverse tasks.


Here is a list of VC criteria and types of examples we have (Q: Question, GT: Ground Truth, Pred: Prediction):

- Short-form QA with ground truth < 3 words:
```
Q: What language(s) is/are represented in the dataset?
GT: English
Pred: English , German , French , Greek .
```

VC criteria (used in the paper): containing more than 3 words

- Short-form QA with ground truth > 3 words:
```
Q: What controversial topics are experimented with?
GT: political events such as elections , corruption cases or justice decisions
Pred: The controversial topics experimented with include the Israeli-Palestinian conflict , the 2016 US presidential election , the Bolsonaro presidential campaign , the Ukraine-Russia conflict , and the Netanyahu corruption case .
```
VC criteria: containing more tokens than ground truth:



- Chain-of-Thought:
```
Q: An astronaut lands on a planet whose mass and radius are each twice that of Earth. If the astronaut weighs 800 N on Earth, how much will he weigh on this planet?
GT: 400 N
Pred:
Here's the solution:

1. Astronaut's weight on Earth is 800 N.
2. Planet's mass is twice Earth's, so astronaut's weight on planet is 2 × 800 N = 1600 N.
3. Answer: 1600 N. 0.5 0.3333333333333333
```

---

> ### Author Response · Authors · 2024-12-04
> **Summary of Discussion (Part 2)**
>
> VC criteria (used in the paper):  Any step contains more than k words. k is the maximum length of the ground truth steps. Given that MMLU does not provide reasoning steps as ground truth, we assume that the steps can always be decomposed into sub-steps with less than k=10 words, and ask the model to generate each step in less than 10 words.
>
> - Planning:
> ```
> Q: Find a trip plan of visiting the cities for 25 days by taking direct flights to commute between them.
> GT:
> Day 1-2: Arriving in Barcelona for 2 days.
> Day 2: Fly from Barcelona to Prague.
> Day 2-5: Visit Prague for 4 days.
> Day 5: Fly from Prague to Edinburgh.
> …
> Pred:
> Step 1: Start in Bucharest for 5 days.
> Step 2: Take a direct flight to Lyon for 4 days.
> Step 3: Take a direct flight to Barcelona for 2 days, attending the workshop.
> …
> ```
>
> VC criteria: Any step contains more than k words. k is the maximum length of the ground truth steps.
>
> > ### **Other concerns and modifications of the PDF**
>
> We list the modifications in the PDF as the summary of the discussion on how we addressed other concerns since we added the discussion results to the PDF:
>
> Main manuscript
>
> - [3ZZU] (line 167) Clarify that $y$ is optional for the verbosity detector.
> - [3ZZU] (line 158) Clarify the meaning of $|r|$
> - [3ZZU] (line 365) Fix the typo of llama-3-70B
> - [Tbb5] (Section 3.2) Reverse the minuend and subtrahend in equation 3.2
> - [esHV] We rewrite the abstract and introduction to define the research problem more clearly and emphasize its significance.
>
> Supplementary Experiments
>
> - [Tbb5] (Appendix C.1) Comparison with Uncertainty-based Routing Algorithm (Figure 7)
> - [Tbb5, esHV] (Appendix C.2) Verbosity Compensation in Trip Planning Dataset (Table 11)
> - [esHV] (Appendix C.3) Robustness of Verbosity Compensation against Prompt Choices (Table 12)
> - [Tbb5] (Appendix C.4) Evaluation of Verbosity and Performance on Same Test Instances (Table 13)
> - [esHV] (Appendix C.5) Latency Comparison of CaSel Algorithm and Individual Models (Table 14)
> - [esHV] (Appendix C.6)The influence of the digits in responses (Table 15)
> - [Tbb5, esHV] (Appendix C.7) Response Length of Chain-of-Thought Experiments  (Table 16)
>
>
> Finally, we deeply thank the efforts of all reviewers in the whole process that greatly enhanced our paper. We will further carefully incorporate the new discussion after the PDF  submission deadline into the final version.

---

### Meta-Review · Area_Chair_eJEm · 2024-12-19

**Metareview:**

This paper introduces an under-explored unexpected behavior of LLMs, named as "Verbosity", reflecting the extent of uncertainty or hesitation when providing responses. To evaluate verbosity, the authors propose "Verbosity Compensation" measured by the difference between the score of concise response and verbose response. Experiments demonstrate that existing LLMs suffer from the verbosity issue. A cascade model selection algorithm is then proposed to mitigate this problem.

Strengths:
- The introduction and formalization of the verbosity problem is novel and interesting.
- A novel metric is proposed to quantify the level of verbosity. Analysis reveals a strong connection between verbosity compensation and uncertainty, suggesting a potential direction to enhance LLM capabilities.
- Extensive experiments are conducted showcasing that LLMs indeed suffer significantly from the verbosity issue. A simple method is given to mitigate this issue.

Weaknesses:
- The paper lacks clarity. It is not clear from Section 3 how a response is compressed with fewer tokens while keeping the same meaning, i.e., how to derive $V(x,y,r)$.  It is also not clear how it works when the ground-truth answer is unavailable, leaving it difficult to understand how to make cascade model selection free of reference. (Although the authors provided more explanations during rebuttal, the current paper does not make it very clear.)
- The paper emphasizes its distinction from traditional way of looking at the response length. However, it uses length of the response to derive V (e.g., larger than 3 tokens).
- The experiments were originally conducted over a limited selection of datasets where the gold answer should be within 3 tokens. This limits the applicability of this method. The authors provided more diverse selection strategies during the rebuttal phase, but it is not fully convincing that the number of tokens or steps could indicate the verbosity label.

**Additional Comments On Reviewer Discussion:**

- Reviewer concerned about the fairness of using different datasets to compute VC and the authors conducted additional experiments to validate this point.
- Additional experiments on more datasets not limited to the answer length (which is 3) are raised. In response, the authors conducted addtional experiments to address this question.
- Latency analysis of the proposed mitigation strategy was raised and addressed by the authors.

Overall, I appreciate the authors' effort in conducting additional experiments to address the reviewers' concerns. Despite the effort, the reviewers remain concerned about the clarity and scope which could be further enhanced to reach a publishable state.

---

### Decision · Program_Chairs · 2025-01-22

Reject